# Inhibition of the proton-activated chloride channel PAC by PIP$_2$

**Ljubica Mihaljević[1†], Zheng Ruan[2†], James Osei-Owusu[1], Wei Lü[2]\*, Zhaozhu Qiu[1,3]\***

[1]Department of Physiology, Johns Hopkins University School of Medicine, Baltimore, United States; [2]Department of Structural Biology, Van Andel Institute, Grand Rapids, United States; [3]Solomon H. Snyder Department of Neuroscience, Johns Hopkins University School of Medicine, Baltimore, United States

**Abstract** Proton-activated chloride (PAC) channel is a ubiquitously expressed pH-sensing ion channel, encoded by *PACC1* (*TMEM206*). PAC regulates endosomal acidification and macropinosome shrinkage by releasing chloride from the organelle lumens. It is also found at the cell surface, where it is activated under pathological conditions related to acidosis and contributes to acid-induced cell death. However, the pharmacology of the PAC channel is poorly understood. Here, we report that phosphatidylinositol (4,5)-bisphosphate (PIP$_2$) potently inhibits PAC channel activity. We solved the cryo-electron microscopy structure of PAC with PIP$_2$ at pH 4.0 and identified its putative binding site, which, surprisingly, locates on the extracellular side of the transmembrane domain (TMD). While the overall conformation resembles the previously resolved PAC structure in the desensitized state, the TMD undergoes remodeling upon PIP$_2$-binding. Structural and electrophysiological analyses suggest that PIP$_2$ inhibits the PAC channel by stabilizing the channel in a desensitized-like conformation. Our findings identify PIP$_2$ as a new pharmacological tool for the PAC channel and lay the foundation for future drug discovery targeting this channel.

**\*For correspondence:**
wei.lu@vai.org (WL);
zhaozhu@jhmi.edu (ZQ)

[†]These authors contributed equally to this work

**Competing interest:** The authors declare that no competing interests exist.

## Editor's evaluation

The recently identified Proton-Activated Chloride (PAC) channel is ubiquitously expressed and has important roles in intracellular organelles and its function in the plasma membrane is associated with human pathologies. Combining electrophysiology, site-directed mutagenesis, lipid pharmacology, and single particle cryo-electron microscopy, this valuable study provides solid evidence to identify a site on the extracellular half of the transmembrane domain of PAC channels that could be occupied by PIP2 and related lipids to promote channel desensitization. These findings are relevant because pharmacological information for this important ion channel is absent.

## Introduction

Proton-activated chloride channel PAC (also known as acid-sensitive outwardly rectifying anion channel or ASOR) is an evolutionarily conserved membrane protein with ubiquitous expression across different tissues. Since the recent discovery of its molecular identity (*Yang et al., 2019*; *Ullrich et al., 2019*), PAC has been implicated in important biological functions, such as endosomal trafficking and macropinocytosis (*Osei-Owusu et al., 2021*; *Zeziulia et al., 2022*). In the endosome, low luminal pH activates PAC to mediate chloride efflux from the lumen. Thereby, PAC actively regulates luminal acidification by depleting the counter ion, chloride, and preventing proton accumulation in the endosome (*Osei-Owusu et al., 2021*). During macropinocytosis PAC mediates the shrinkage of macropinosomes by releasing chloride into the cytoplasm (*Zeziulia et al., 2022*). In addition to localizing to the intracellular organelles, PAC also traffics to the plasma membrane, where it is involved in several pathological

conditions associated with acidosis. For example, upon ischemic stroke, PAC is activated by drops in tissue pH, allowing the entry of chloride into the cells. This subsequently causes cellular swelling and contributes to acid-induced brain injury (*Osei-Owusu et al., 2020*; *Yang et al., 2019*).

PAC is a homotrimer that forms a chloride-selective pore in the membrane, and it senses changes in pH via its large extracellular domain (ECD) (*Deng et al., 2021*; *Osei-Owusu et al., 2022a*; *Ruan et al., 2020*; *Wang et al., 2022*). PAC channel is closed at neutral pH and becomes activated when the pH drops below 5.5 (*Yang et al., 2019*). The proton binding to the ECD is directly coupled with the channel opening in the transmembrane domain (TMD) (*Osei-Owusu et al., 2022a*). After prolonged exposure to pH 4.6 or below, the PAC channel slowly desensitizes (*Osei-Owusu et al., 2022b*). The desensitization of PAC is pH-dependentthat is, under more acidic conditions, desensitization is stronger (*Osei-Owusu et al., 2022b*). This is regulated by several key residues localized at the ECD–TMD (*Osei-Owusu et al., 2022b*). For example, the E94R mutant displays fast desensitization even at pH 5.0, when the wild-type channel does not exhibit obvious current decay (*Osei-Owusu et al., 2022b*). In addition to the resting and desensitized structures (*Ruan et al., 2020*), an open conformation of PAC was recently reported (*Wang et al., 2022*). The transition between closed, open and desensitized PAC channel conformations involves major structural rearrangements inside the lipid bilayer (*Ruan et al., 2020*; *Wang et al., 2022*). It is now widely accepted that the membrane shapes ion channel function and structure. Lipid composition and the thickness of the membrane can directly control or fine-tune the gating of certain ion channels (*Rosenhouse-Dantsker et al., 2012*). However, whether the PAC channel is regulated by lipids is unknown.

The most common and best-studied lipid regulator of ion channel function is phosphatidylinositol (4,5)-bisphosphate (PIP$_2$). PIP$_2$ is a negatively charged phospholipid, predominantly found in the inner leaflet of the plasma membrane (*Suh and Hille, 2008*), with few reports that a small amount can localize to the outer leaflet as well (*Gulshan et al., 2016*; *Yoneda et al., 2020*). Although it accounts for less than 1% of the total phospholipids in the plasma membrane, it is a principal signaling molecule and an essential cofactor for ion channel function (*Hansen, 2015*; *Suh and Hille, 2008*). PIP$_2$ binds to ion channels directly and modulates their function by facilitating channel opening, preventing current rundown/desensitization, or inhibiting channel activity (*Gada and Logothetis, 2022*; *Suh and Hille, 2008*). At least 10 different ion channel families are dependent on PIP$_2$ for their activity (*Gada and Logothetis, 2022*; *Suh and Hille, 2008*). A handful of ion channles are negatively regulated by PIP$_2$ (*Gada and Logothetis, 2022*; *Suh and Hille, 2008*), one of which is TMEM16B– the only chloride channel that is reported to be inhibited in the presence of PIP$_2$ to this date (*Ta et al., 2017*).

Major breakthroughs have been made in recent years to characterize the structure and function of the PAC channel in biology and disease, but its regulation by endogenous molecules remains largely unexplored. PAC recently emerged as a target of interest for acidosis-related diseases, and much progress has been made to identify natural and synthetic compounds that can inhibit PAC channel (*Figueroa, 2020*; *Okada et al., 2021*; *Sato-Numata et al., 2016*). DIDS (4,4-diisothiocyanatostilbene -2,2-disulfonic acid), a broad-spectrum chloride channel blocker, with a half-maximal inhibition (IC$_{50}$) of 2.9 μM in HEK293 cells (*Okada et al., 2021*) is the most commonly used PAC inhibitor. To date, arachidonic acid (IC$_{50}$ of 8.9 μM; *Okada et al., 2021*) and pregnenolone sulfate (IC$_{50}$ of 10 μM; *Figueroa, 2020*) are the only reported potential biological inhibitors of the acid-induced chloride currents ($I_{Cl,H}$) mediated by PAC, yet their mechanism and binding sites are not fully characterized. Here, we show that PIP$_2$ potently inhibits PAC and present the first structure of PAC channel with an inhibitor. Furthermore, we elucidate the molecular mechanism by which PIP$_2$ inhibits PAC by stabilizing a desensitized-like conformation of the channel.

## Results

### PIP$_2$ inhibits PAC channel activity

Considering the widespread influence of PIP$_2$ on ion channel function, we hypothesized that PIP$_2$ could potentially regulate the PAC channel. To test whether PIP$_2$ modulates PAC activity, we applied a soluble version of PIP$_2$ lipid, dioctanoyl phosphatidylinositol 4,5 bisphosphate (diC$_8$-PIP$_2$), to HEK293 cells, which endogenously express the acid-induced chloride currents. Whole-cell $I_{Cl,H}$ were detected in real-time, by perfusing the cells with an acidic solution at pH 5.0, followed by an application of 10 μM diC$_8$-PIP$_2$. Immediately upon adding PIP$_2$, there was a rapid drop in $I_{Cl,H}$, followed by a steady

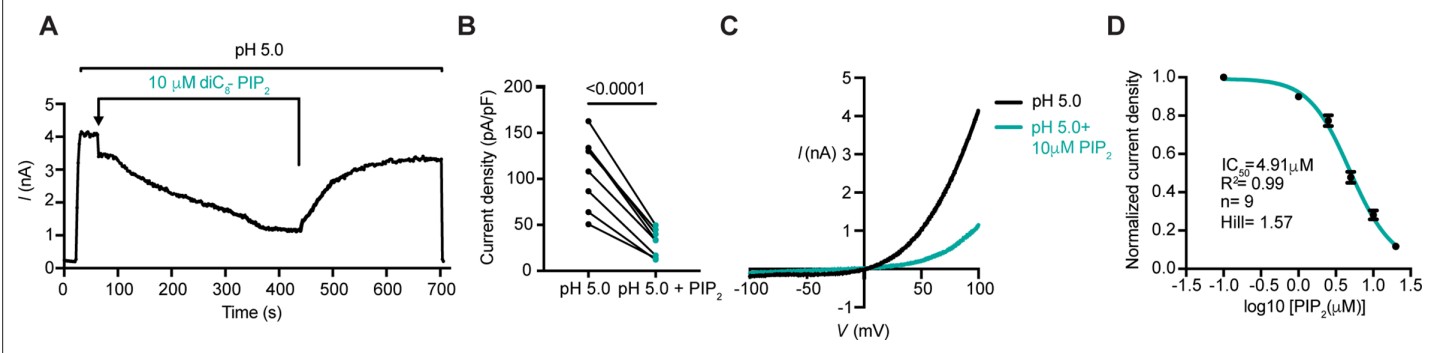

**Figure 1.** $PIP_2$ inhibits the PAC channel activity. (**A**) Representative whole-cell current trace at + 100 mV (2 s/sweep) showing inhibition of endogenous PAC currents by bath perfusion of soluble $diC_8$-$PIP_2$ at 10 μM concentration. (**B**) PAC current densities before and after application of $PIP_2$ for 150 s. Statistical significance was determined using a two-tailed Student's paired *t*-test. (**C**) Representative *I/V* curve of pH 5-induced PAC currents before and after $PIP_2$ treatment. (**D**) Dose-dependent inhibition of pH 5-induced PAC currents by $PIP_2$ yielded a half-maximal inhibition, $IC_{50}$, of 4.91 μM with a Hill slope of 1.57. Bars are reported as mean ± SEM.

The online version of this article includes the following source data and figure supplement(s) for figure 1:

**Source data 1.** Data and statistics plotted in *Figure 1*.

**Figure supplement 1.** $PIP_2$ does not bind to the closed PAC channel.

**Figure supplement 1—source data 1.** Data and statistics plotted in *Figure 1—figure supplement 1*.

decline (*Figure 1A*). Furthermore, this effect was reversible by washing out the soluble lipid from the cell membrane with a pH 5.0 solution (*Figure 1A*). Approximately 37% of the initial current amplitude was detectable after $PIP_2$ perfusion for 150 s (*Figure 1B and C*). This short timescale of $PIP_2$ action on PAC activity indicates that its effect is most likely direct. Further supporting this, $PIP_2$ inhibited $I_{Cl,H}$ in a dose-dependent manner, with half-maximal inhibition, $IC_{50}$, of 4.9 μM (*Figure 1D*).

At neutral pH, the PAC channel is in its resting/closed state, and it is activated/open when the pH drops below 5.5 at room temperature (*Yang et al., 2019*). To examine if $PIP_2$ exerts its effect on the PAC channel in its closed or open state, we pre-treated the cells with soluble $PIP_2$ at pH 7.3, and then activated the channel with acid. $I_{Cl,H}$ amplitude at pH 5.0 did not show any significant difference before and after perfusion of $PIP_2$ at the neutral pH (*Figure 1—figure supplement 1A, B*). This result suggests that $PIP_2$ may not act on the resting state of PAC and is only effective once the channel undergoes proton-induced activation or the subsequent desensitization.

## Phosphates and the acyl chain synergistically contribute to $PIP_2$-mediated PAC inhibition

To test if there is a preference among different phosphatidylinositol lipids, we used soluble ($diC_8$) versions of lipids at 10 μM concentration and compared their inhibitory effects on pH 5.0-induced endogenous PAC currents. PI(3)P (phosphatidylinositol 3-phosphate) with a single phosphate on its inositol headgroup, inhibited PAC significantly less than either bisphosphonates, PI(4,5)$P_2$ or PI(3,5)$P_2$ (phosphatidylinositol (3,5)-bisphosphate) (*Figure 2A*). The additional phosphate on $PIP_3$ (phosphatidylinositol (3,4,5)-trisphosphate) yielded the $IC_{50}$ of 3 μM (*Figure 2B*). Therefore, to reach potent inhibition, a minimum of two phosphates on the inositol headgroup are required. This is additionally supported by a modest inhibitory effect of PI (phosphatidylinositol) that does not have any active headgroup phosphates (*Figure 2C*). Interestingly, IP3 (inositol 1,4,5-trisphosphate), a triple-phosphorylated inositol headgroup without an acyl chain, displayed a similarly modest inhibition on PAC as PI (*Figure 2D*). Phosphates on the headgroup are therefore necessary, but not sufficient for PAC inhibition, indicating that the lipid chain contributes to inhibitory properties of $PIP_2$ as well. $diC_8$-diacyl-glycerol (DAG), the lipid chain without inositol head, had no inhibitory effect on PAC (*Figure 2D*). Acyl chain alone is therefore not sufficient to inhibit PAC.

Next, we examined the inhibitory effect of phosphatidylinositol lipids with varying chain lengths. $PIP_2$ with two carbons less on its acyl chain, $diC_6$-$PIP_2$, was significantly less potent than $diC_8$-$PIP_2$ in inhibiting $I_{Cl,H}$ (*Figure 2D*). On the other hand, full-length $diC_{18:0}$-$_{20:0}$ -$PIP_2$, displayed a potent inhibition on the PAC channel, comparable to that of $diC_8$-$PIP_2$ (*Figure 2D*). Based on these results, we conclude

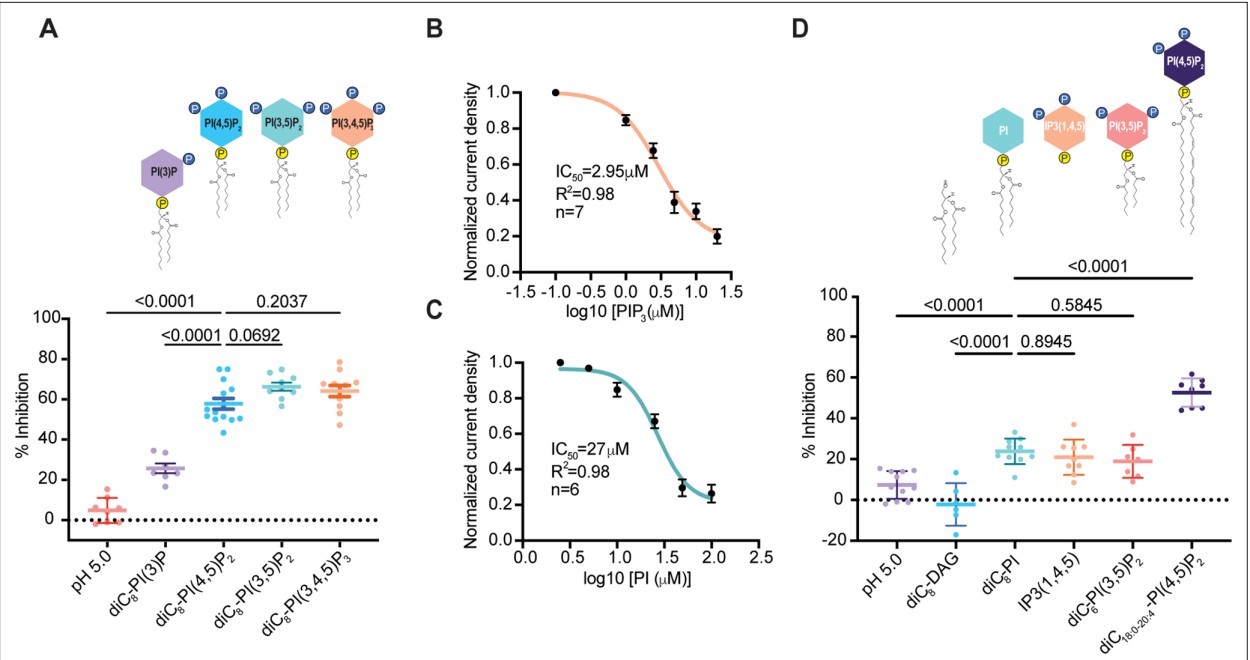

**Figure 2.** Phosphates and acyl chain length synergistically contribute to PAC inhibition by PIP$_2$. (**A**) Percent inhibition of pH 5-induced PAC currents by different diC$_8$-phosphatidylinositol lipids at 10 µM concentration: PI(3)P, PI(4,5)P$_2$, PI(3,5)P$_2$, and PI(3,4,5)P$_3$. Statistical significance was determined using ordinary one-way ANOVA with the Dunnett post hoc test. Bars are reported as mean ± SEM. (**B**) Dose-dependent inhibition of PAC currents by PIP$_3$. Bars are reported as mean ± SEM. (**C**) Dose-dependent inhibition of PAC currents by phosphatidylinositol (PI). Bars are reported as mean ± SEM. (**D**) Percent inhibition of pH 5-induced PAC currents by phosphatidylinositol lipids of different acyl chain length 10 µM concentration: Diacylglycerol (DAG), diC$_8$-PI, IP3(1,4,5), diC$_6$-PI(3,5)P$_2$ and diC$_{18:0-20:4}$-PI(4,5)P$_2$. Statistical significance was assessed using ordinary one-way ANOVA with the Dunnett post hoc test. Bars are reported as mean ± SEM.

The online version of this article includes the following source data for figure 2:

**Source data 1.** Data and statistics plotted in **Figure 2**.

that an acyl chain with a minimum of 8 carbons is required for potent inhibition of PAC by PIP$_2$. Together, the number of phosphates on the inositol headgroup and lipid chain length synergistically contribute to the inhibitory potency of PIP$_2$ to the PAC channel.

## Cryo-EM structure reveals the PIP$_2$ binding site on the PAC channel

PIP$_2$ often binds to motifs on the intracellular side of ion channels, which contain positively charged residues that directly interact with the negatively charged phosphates on its inositol head. Indeed, the desensitized structure of PAC that we determined previously, a cluster of lipid-like density is present in the cytoplasmic fenestration area (**Ruan et al., 2020**). The C-terminus of TM2 has several positively charged residues, including K325, K329, K333, R335, K336, R337, K340, R341, R342, that we focused on initially and studied for their impact on PIP$_2$ sensitivity (**Figure 3—figure supplement 1A**). Mutating single or triple lysine and arginine residues to alanine or making a 10-residue deletion at the C-terminal domain did not affect PIP$_2$-mediated inhibition on pH 5.0-induced PAC currents (**Figure 3—figure supplement 1B**). In addition, when diC$_8$-PIP$_2$ was applied to the cells through an intracellular solution in the patch pipette, at a physiological concentration of 10 µM, there was no detectable change in the endogenous PAC current amplitude (**Figure 3—figure supplement 1C**). Similarly, $I_{Cl,H}$ remained intact when endogenous PIP$_2$ was depleted from the inner membrane leaflet using 100 µg/ml Poly-L-Lysine (PLL) in the patch pipette (**Figure 3—figure supplement 1D**). These results are surprising because endogenous PIP$_2$ is known to be almost exclusively localized to the inner leaflet of the plasma membrane. Thus, the effect we observed with the perfusion of exogenous PIP$_2$ may occur via inhibition of the PAC channel through a potentially unconventional mechanism.

To reveal the mechanism underlying PIP$_2$ inhibition, we solved the cryo-EM structure of PAC in nanodiscs with 0.5 mM diC$_8$-PIP$_2$ at pH 4.0 to an overall resolution of 2.70 Å (**Figure 3A**, **Supplementary file 1**). The structure adopts a conformation similar to the previously reported desensitized state

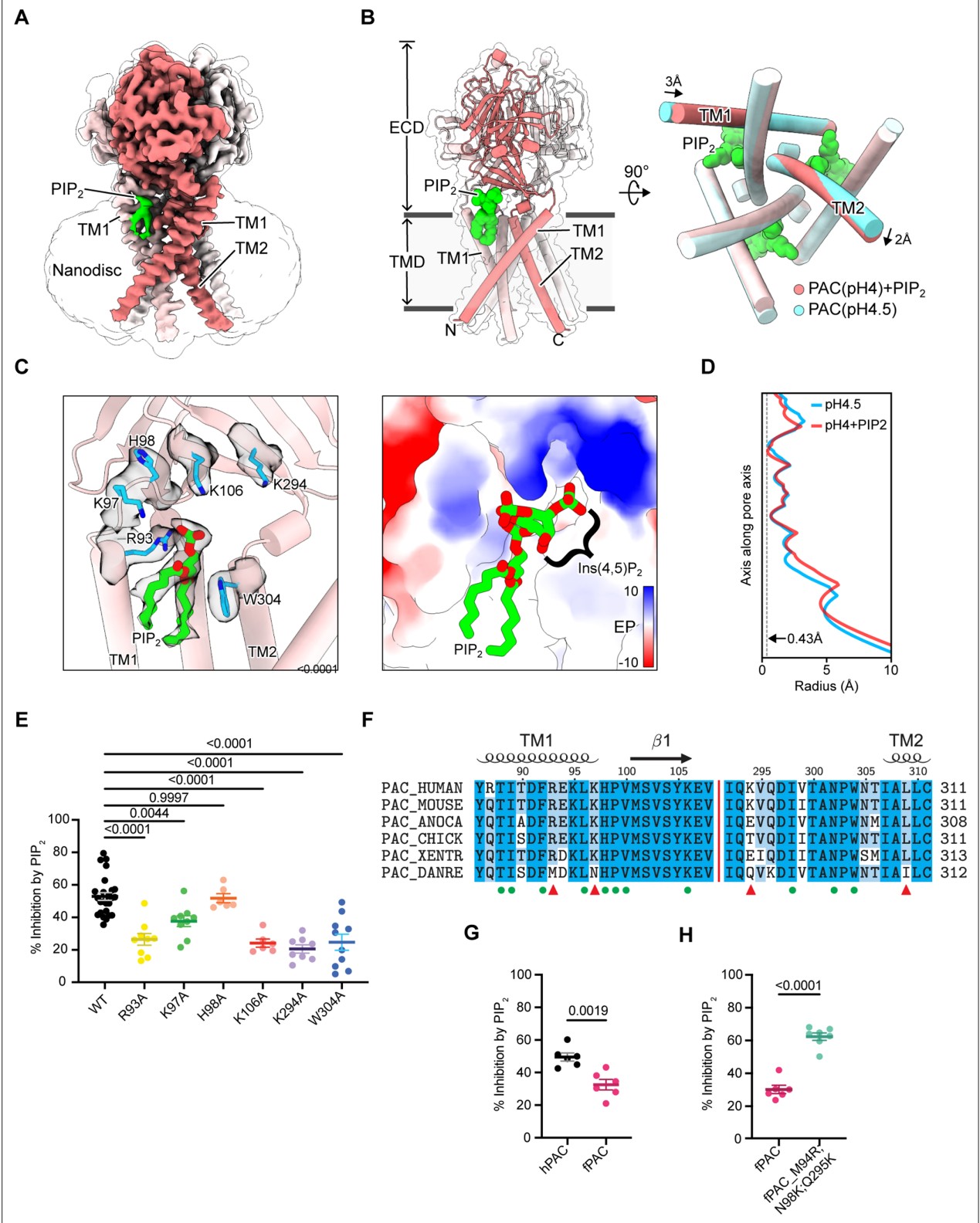

**Figure 3.** PIP$_2$ binds directly to the PAC channel. (**A**) Cryo-EM structure of PAC channel at pH 4.0 with bound PIP$_2$. One subunit is shown in red with TM1 and TM2 labeled. The density corresponding to putative PIP$_2$ is colored green. (**B**) The structural model of PAC channel at pH 4.0 with bound PIP$_2$ in side view (left) and bottom-up view (right). For comparison, the PAC channel at pH 4.5 without PIP$_2$ (PDBID: 7SQH) is shown in cyan for the bottom-up view (right). (**C**) A close-up view of the PIP$_2$ binding site. (Left) Cartoon representation of the PIP$_2$ binding site. Important residues relevant to the study of the

*Figure 3 continued*

putative PIP$_2$ binding site, including R93, K97, H98, K106, K294, and W304, are shown in stick. Cryo-EM densities for PIP$_2$ and the nearby residues are shown in a semi-transparent surface. The Ins(4,5)P$_2$ group is not resolved in the Cryo-EM map and thus not modeled in the deposited structure. (Right) Surface representation of the PIP$_2$ binding site colored with electrostatic potential. Unit in kcal/mol/e$^-$. A full PIP$_2$ molecule, including the hypothetically positioned Ins(4,5)P$_2$ group, is shown in the right panel. (**D**) The pore profile of PAC at pH 4.0 with PIP$_2$ (PDBID: 8FBL) and at pH 4.5 without PIP$_2$ (PDBID: 7SQH). The smallest radius along the pore axis is 0.43 Å, suggesting that both structures are impermeable to chloride ions. (**E**) Mutating PIP$_2$-binding residues to alanine significantly decreases diC$_8$-PIP$_2$-mediated inhibition on pH 5-induced PAC currents. The constructs were expressed in PAC KO HEK293 cells for recordings. Statistical significance was assessed using one-way ANOVA with the Dunnett post hoc test. Bars are reported as mean ± SEM. (**F**) Multiple sequence alignment of several PAC orthologs. Key residues that form the PIP$_2$ binding site are labeled using green dots. Binding site residues that are not conserved in zebrafish PAC (PAC_DANRE) are indicated by red triangles. (**G**) Percent inhibition (mean ± SEM) of hPAC or fPAC current at pH 5.0 by 10 µM diC$_8$-PIP$_2$. Zebrafish PAC (fPAC) shows significantly less inhibition by PIP$_2$ compared to human PAC. Statistical significance was determined using a two-tailed Student's unpaired *t*-test. (**H**) Mutating zebrafish PAC residues to the corresponding human PAC residues, M94R, N98K and Q295K, significantly increases the inhibition by PIP$_2$ in comparison to the wild-type zebrafish PAC. Statistical significance was determined using a two-tailed Student's unpaired *t*-test. Bars are reported as mean ± SEM.

The online version of this article includes the following source data and figure supplement(s) for figure 3:

**Source data 1.** Data and statistics plotted in *Figure 3*.

**Figure supplement 1.** The binding site for PIP$_2$ is not on the intracellular side of the PAC channel.

**Figure supplement 1—source data 1.** Data and statistics plotted in *Figure 3—figure supplement 1*.

**Figure supplement 2.** The cryo-EM data processing workflow of human PAC in nanodisc at pH 4.0 with 0.5 mM PIP$_2$ dataset.

**Figure supplement 3.** The reconstruction metrics of human PAC in nanodisc at pH 4.0 with 0.5 mM PIP$_2$.

**Figure supplement 4.** Putative PIP$_2$-binding residues mapped in the PAC structures in the resting, open, and desensitized states, respectively.

at low pH (*Ruan et al., 2020*). However, a strong branched lipid density is observed on the cryo-EM map of the PAC channel in the outer membrane leaflet, between TM1 and TM2 of adjacent subunits (*Figure 3A*, *Figure 3—figure supplement 2*, *Figure 3—figure supplement 3*). We suspected that the density may represent a bound PIP$_2$ molecule, although we cannot rule out the possibility that this density may represent other types of lipids, such as phosphatidic acid. When trying to fit a diC$_8$-PIP$_2$ molecule into this density (*Figure 3B*), we found that the phosphatidyl group is reasonably well defined, with its phosphate group forming a salt bridge interaction with R93 and its two acyl tails interacting with a number of hydrophobic residues on both transmembrane helices, including W304 (*Figure 3C*). In contrast, the hydrophilic head group, such as inositol-4,5-bisphosphate (Ins(4,5)P$_2$) moiety of PIP$_2$, is not resolved and therefore not modeled. The absence of the head group may be explained by its intrinsic flexibility and/or, if it is Ins(4,5)P$_2$, its susceptibility to radiation damage due to the negative charges it carries. Nevertheless, the local biochemical environment of the site is consistent with PIP$_2$ binding, in which the hypothetical location of Ins(4,5)P$_2$ is surrounded by several positively charged residues, including K97, K106, and K294 (*Figure 3C*). The putative PIP$_2$ binding site is also in accordance with our observation that a higher number of negatively charged phosphates, as well as the presence of an acyl chain, contribute to stronger channel inhibition by PIP$_2$ (*Figure 2*). Moreover, small, but notable, conformational changes are observed in the transmembrane helices (TM1 and TM2) upon adding PIP$_2$, which supports the binding of PIP$_2$ to PAC. Specifically, TM1 tilts inside by 3 Å which causes a concerted rotation motion of TM2 (*Figure 3B*). The pore radius profile is similar to the desensitized state of PAC without PIP$_2$, with the smallest radius of 0.43 Å (*Figure 3D*). Therefore, the PIP$_2$-bound conformation also represents a non-conductive state (*Figure 3A–C*).

To validate our structural model and the putative PIP$_2$ binding site, we carried out site-directed mutagenesis and patch-clamp electrophysiological experiments. When overexpressed in *PAC* knockout (KO) HEK293 cells, mutation of any residues R93, K97, K106, K294, and W304 to alanine significantly relieved the inhibition by PIP$_2$ on pH 5.0-induced PAC currents, confirming that this was indeed its binding site on the channel (*Figure 3E*). We also examined an adjacent residue, H98, which is not at a distance from the binding site that would allow direct interaction with PIP$_2$. As expected, H98A mutant was still sensitive to PIP$_2$ inhibition (*Figure 3E*), suggesting that PIP$_2$ specifically recognizes the binding pocket observed in our structure. None of the mutations we tested affected the PAC channel activity, as indicated by the normal current densities and very small desensitization at pH 5.0 (*Figure 3—figure supplement 1E, F*). Moreover, the PIP$_2$ binding site is unlikely to exist in the resting and activated states of PAC because the TMD, particularly TM1, undergoes significant conformational

changes relative to the desensitized state, and therefore a PIP$_2$ molecule cannot be accommodated in these states (*Figure 3—figure supplement 4*).

Putative PAC PIP$_2$-binding residues are conserved amongst higher vertebrates. On the contrary, in zebrafish (*Danio rerio*), several PIP$_2$-binding site residues are different, including M94, N98, and Q295 (*Figure 3F*). Interestingly, the zebrafish PAC channel (fPAC) was significantly less inhibited by PIP$_2$ compared to the human PAC (hPAC) when overexpressed in *PAC* KO HEK293 cells (*Figure 3G*). To test if the reduced PIP$_2$ sensitivity of zebrafish PAC is due to these amino acid differences, we used site-directed mutagenesis to convert zebrafish residues to the corresponding ones of the human PAC channel. Interestingly, zebrafish triple mutant M94R, N98K, Q295K (fPAC numbering) showed a significant increase in PIP$_2$ inhibition when compared to the wild-type channel (*Figure 3H*, *Figure 3—figure supplement 1G*). This further substantiates our finding that the PIP$_2$ binding site on the PAC channel is located in the outer membrane leaflet.

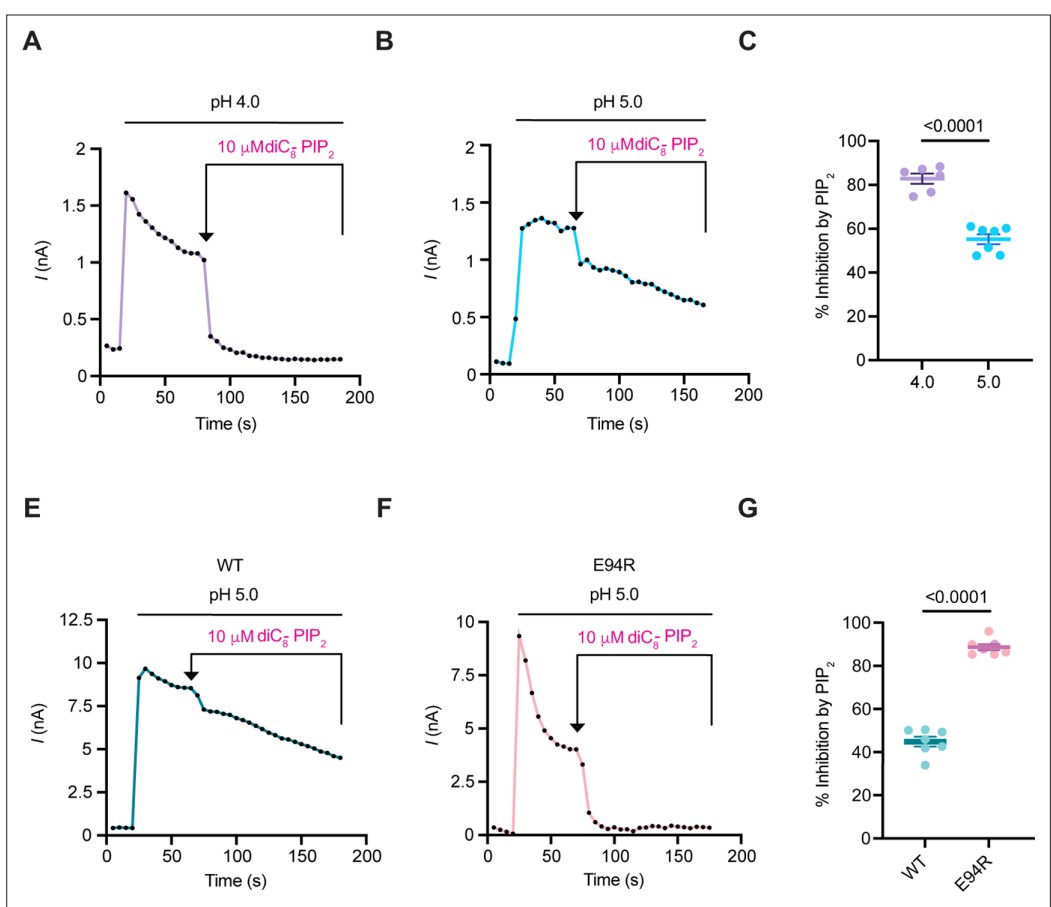

**Figure 4.** PIP$_2$-mediated PAC inhibition correlates with the degree of channel desensitization. (**A**, **B**) Representative current traces at + 100 mV (5 s/sweep) of endogenous PAC currents at pH 4.0 and 5.0 treated with 10 µM diC$_8$-PIP$_2$. diC$_8$-PIP$_2$ was applied after desensitized current reached a plateau. (**C**) Percent inhibition (mean ± SEM) of PAC currents at pH 4.0 and 5.0, 100 s after perfusion of 10 µM diC$_8$-PIP$_2$. Statistical significance was determined using a two-tailed Student's unpaired *t-test*. (**D**, **E**) Representative current traces at + 100 mV (5 s/sweep) of overexpressing PAC WT and E94R at pH 5.0 treated with 10 µM diC$_8$-PIP$_2$. (**F**) Percent inhibition (mean ± SEM) of PAC WT and E94R currents at pH 5.0, 100 s after perfusion of 10 µM diC$_8$-PIP$_2$. Statistical significance was determined using a two-tailed Student's unpaired *t-test*.

The online version of this article includes the following source data for figure 4:

**Source data 1.** Data and statistics plotted in *Figure 4*.

## PIP$_2$-mediated PAC inhibition correlates with the degree of channel desensitization

Since PIP$_2$-bound PAC structure resembles the desensitized state and the PAC channel exhibits apparent desensitization at pH 4.0 (*Osei-Owusu et al., 2022b*) we sought to examine how pH may influence PIP$_2$-mediated PAC inhibition by applying diC$_8$-PIP$_2$ at pH 4.0 when the currents stabilized after the initial fast desensitization (*Figure 4A and B*). The percentage of PAC inhibition by PIP$_2$ increased significantly at pH 4.0 compared with pH 5.0 (*Figure 4C*). These results suggest that PIP$_2$ inhibition is more effective when the PAC channel is already poised toward the desensitized state under more acidic conditions.

We recently showed that reversing the charge of E94 residue to E94R induces PAC desensitization, even at pH 5.0 (*Figure 4D and E*; *Osei-Owusu et al., 2022b*). Structurally, E94 is located in TM1, facing the opposite side of the PIP$_2$ binding pocket. Therefore, the E94 mutation is unlikely to affect PIP$_2$ binding directly, representing an ideal candidate to test if there is a correlation between PIP$_2$ inhibition and channel desensitization. Indeed, we found that PIP$_2$ exerted a much higher degree of inhibition on the E94R mutant than the WT PAC channel (*Figure 4F*). Because E94 is distal to the PIP$_2$ binding site, these effects are most likely due to the altered conformational dynamics toward desensitization. It is also important to note that the desensitization of PIP$_2$-binding mutants was not altered (*Figure 3—figure supplement 1F*). Together with its pH-dependency, our data suggest that PIP$_2$ inhibition is more effective when the PAC channel is more prone to becoming desensitized.

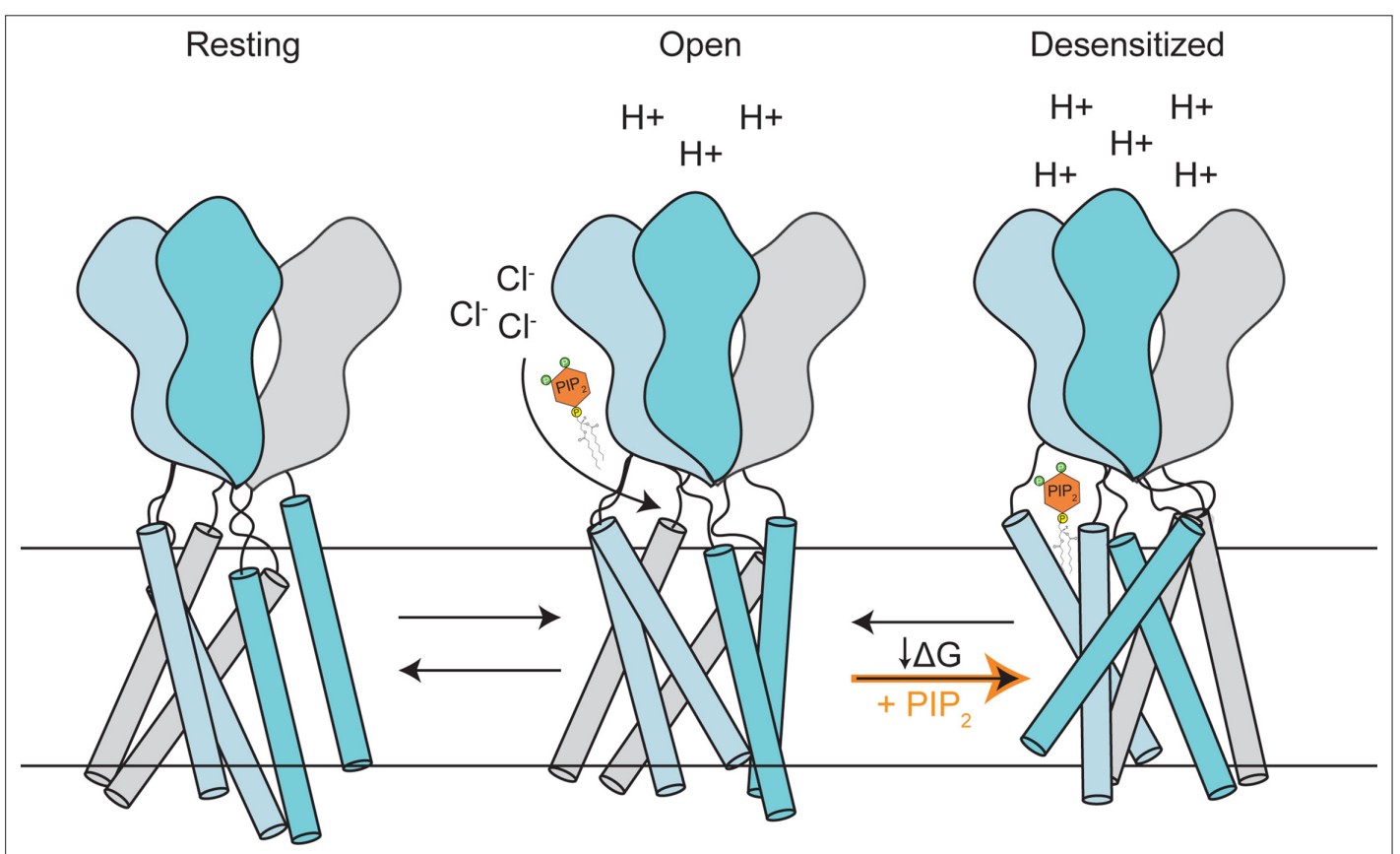

**Figure 5.** A proposed model of PAC inhibition by PIP$_2$. PAC channel adopts resting/open/desensitized states depending on the acidity of the environment. PIP$_2$ selectively binds and stabilizes the desensitized conformation of PAC on the extracellular side of the membrane, altering the conformational/free energy landscape of the channel. As a result, in the presence of PIP$_2$, a significant portion of PAC will be restricted in the desensitized conformation, leading to channel inhibition.

# Discussion

PAC is a novel chloride channel, and its pharmacology is still poorly studied. Here, we showed that $PIP_2$ binds to and potently inhibits the PAC channel with an $IC_{50}$ of ~4.9 μM. This value is comparable to the $EC_{50}$ of other well-studied $PIP_2$-activated ion channels, such as TMEM16A chloride channel (~3.95 μM) and $K_{ir}$ inward rectifying potassium channels (~4.6 μM) (*Le et al., 2019*; *Lopes et al., 2002*). Additionally, half-maximal inhibition of $PIP_2$ binding to PAC seems to be an order of magnitude lower than the $IC_{50}$ (~46 μM) of TMEM16B− the only other chloride channel known to be inhibited by $PIP_2$ prior to this study (*Ta et al., 2017*), although we noticed that this study also reported a relatively high $EC_{50}$ (~53 μM) for TMEM16A (*Ta et al., 2017*). It is worth noting that the Hill coefficient of $PIP_2$-mediated inhibition is estimated to be 1.57, suggesting that a cooperative mode of binding for $PIP_2$ (*Figure 1D*). Our structural analysis further showed that the $PIP_2$ binding site in PAC is located on the extracellular side of the TMD, unlike other ion channels known to be regulated by $PIP_2$, which bind $PIP_2$ on the intracellular side of the TMD.

Our data indicate that the degree of $PIP_2$-mediated PAC inhibition correlates with channel desensitization. The prevalence of desensitized PAC state at pH 4.0 (*Figure 4A*) and in the E94R mutant (*Figure 4F*) facilitates the inhibitory effect of $PIP_2$ on PAC. $PIP_2$-mediated PAC inhibition under these conditions is more effective likely because the desensitized conformation becomes more prevalent, and the binding sites become more accessible. In contrast, under less acidic conditions most of the channels adopt the open/resting states, resulting in less inhibition by $PIP_2$. Together, our results suggest that $PIP_2$ achieves its inhibitory effect by altering the free energy landscape to favor the desensitized state of the PAC channel (*Figure 5*).

Furthermore, we characterized pharmacological properties that contribute to phosphatidylinositol inhibitory potency on PAC. $PIP_3$ displayed the strongest inhibition of PAC, while $IP_3$ had a negligible effect (*Figure 2*). A higher number of phosphates and a longer acyl chain increase the inhibitory potency of the lipid. This indicates that the negative charge on the inositol head, as well as acyl chain insertion into the membrane, synergistically contribute to the lipid binding and stabilization of the desensitized channel state.

$PIP_2$ is primarily localized in the cytosolic side of the plasma membrane. However, a few studies report that a small portion of $PIP_2$ can be detected on the extracellular side (*Gulshan et al., 2016*; *Yoneda et al., 2020*). In RAW264.7 macrophages and baby hamster kidney (BHK) fibroblasts, ATP-binding cassette transporter A1 (ABCA1) facilitates the redistribution of $PIP_2$ from the inner to the outer leaflet of the plasma membrane (*Gulshan et al., 2016*). Furthermore, in freshly isolated mouse bone marrow cells, $PIP_2$ is localized on the cell surface (*Yoneda et al., 2020*). Binding of PI, and potentially $PIP_2$, in the outer membrane leaflet has previously been observed in the structure of $Na^+/H^+$ exchanger (NHA2), mediating its activity through stabilization of dimer interface, a mechanism distinct from the one described here. Furthermore, the binding of $PIP_2$ to NHA2 is yet to be confirmed in a physiological context (*Matsuoka et al., 2022*).

The $PIP_2$-binding pocket on the extracellular side of the PAC channel is physiologically unusual. It is therefore unlikely, albeit not impossible, that inhibition of PAC by $PIP_2$ occurs in a physiological setting. Considering the recent discovery of PAC and its wide tissue distribution, we speculate that $PIP_2$ could be negatively regulating PAC in the outer leaflet of the plasma membrane in specialized cells, or under certain conditions that are currently unknown to us. For example, the PAC channel localizes to endosomes and macropinosomes of macrophages. The inner membrane of these intracellular organelles is topologically equivalent to the outer leaflet of the plasma membrane. Some pathogens enter the cells via endocytosis or macropinocytosis and escape degradation in these compartments via fusion of their membrane with the membrane of the organelle. The envelope of some pathogens is enriched in $PIP_2$, such as in the human immunodeficiency virus (HIV; *Mücksch et al., 2019*). Therefore, we speculate that $PIP_2$ could potentially be found in the inner membrane of endosomes during fusion with pathogenic membranes, where it inhibits PAC activity and modulates lumen acidification. However, it is also possible that $PIP_2$-binding site on PAC potentially acts as a proxy for another ligand that has yet to be determined.

In conclusion, to our knowledge, $PIP_2$ is the first PAC channel modulator with a characterized binding site and mechanism of action. Although its physiological significance remains elusive, the novel extracellular $PIP_2$-binding pocket for targeted inhibition of PAC can be exploited for the design of PAC inhibitors that do not have to be cell-permeable. Furthermore, we describe pharmacological

properties necessary for PAC inhibition, which include a stable insertion into the membrane and a negative charge that interacts with the positively charged cluster of residues on the pocket. These insights provide a useful tool for the future design of potential therapeutics for acidosis-related diseases implicating the PAC channel.

# Materials and methods

**Key resources table**

| Reagent type (species) or resource | Designation | Source or reference | Identifiers | Additional information |
|---|---|---|---|---|
| Gene (*Homo sapiens*) | hPAC | doi:10.1126/science.aav9739 | NP_060722/Q9 H813 | |
| Gene (*Danio rerio*) | fPAC | doi:10.1126/science.aav9739 | NP_001278691/Q7SY31 | |
| Recombinant DNA reagent | pEGC-hPAC (plasmid) | doi:10.1038/s41586-020-2875-7 | | |
| Recombinant DNA reagent | pIRES2-EGFP-hPAC (plasmid) | doi:10.1126/science.aav9739 | | |
| Recombinant DNA reagent | pIRES2-EGFP-fPAC (plasmid) | This paper | | In the cell culture section of Materials and methods in this paper |
| Recombinant DNA reagent | pEGC-hPAC | doi:10.1038/s41586-020-2875-7 | | |
| Cell line (*Homo sapiens*) | HEK293T | ATCC | Cat#:CRL-3216 | |
| Cell line (*Homo-sapiens*) | tsA-201 | Sigma Aldrich | Cat#: 85120602 | Cell line (*Homo-sapiens*) |
| Cell line (*Homo sapiens*) | PACC1 KO HEK293T | doi:10.1126/science.aav9739 | | |
| Chemical compound, drug | 08:0 PI (1,2-dioctanoyl-sn-glycero-3-phospho-(1'-myo-inositol) (ammonium salt)) | Avanti Polar Lipids | Cat#:850181 P | |
| Chemical compound, drug | 08:0 PI(4,5)P2 (1,2-dioctanoyl-sn-glycero-3-phospho-(1'-myo-inositol-4',5'-bisphosphate) (ammonium salt)) | Avanti Polar Lipids | Cat#:850185 P | |
| Chemical compound, drug | 08:0 PI(3,5)P2 (1,2-dioctanoyl-sn-glycero-3-phospho-(1'-myo-inositol-3',5'-bisphosphate) (ammonium salt)) | Avanti Polar Lipids | Cat#:850184 P | |
| Chemical compound, drug | 08:0 PI(3)P (1,2-dioctanoyl-sn-glycero-3-(phosphoinositol-3-phosphate) (ammonium salt)) | Avanti Polar Lipids | Cat#:850187 P | |
| Chemical compound, drug | 06:0 PI(3,5)P2 (1,2-dihexanoyl-sn-glycero-3-phospho-(1'-myo-inositol-3',5'-bisphosphate) (ammonium salt)) | Avanti Polar Lipids | Cat#:850174 P | |
| Chemical compound, drug | 18:0-20:0- PI(4,5)P2 (1-stearoyl-2-arachidonoyl-sn-glycero-3-phospho-(1'-myo-inositol-4',5'-bisphosphate)) (ammonium salt) | Avanti Polar Lipids | Cat#:850165 P | |
| Chemical compound, drug | IP3(1,4,5) (D-myo-inositol-1,4,5-triphosphate (ammonium salt)) | Avanti Polar Lipids | Cat#:850115 P | |
| Chemical compound, drug | 08:0 DG (1,2-dioctanoyl-sn-glycerol) | Avanti Polar Lipids | Cat#:800800O | |
| Chemical compound, drug | Poly-L-Lysine (PLL) | Sigma-Aldrich | Cat#:26124-78-7 | |
| Commercial assay or kit | Lipofectamine 2000 | Invitrogen | Cat#:11668–019 | |

*Continued on next page*

*Continued*

| Reagent type (species) or resource | Designation | Source or reference | Identifiers | Additional information |
|---|---|---|---|---|
| Commercial assay or kit | QuikChange II XL site-directed mutagenesis | Agilent Technologies | Cat#:200522 | |
| Software, algorithm | Clampfit 10.7 | Molecular devices | | |
| Software, algorithm | GraphPad Prism 9 | GraphPad | | |
| Software, algorithm | Clustal Omega | https://www.ebi.ac.uk/Tools/msa/clustalo/ | | |
| Software, algorithm | Relion | doi:10.7554/eLife.42166 | | |
| Software, algorithm | Cryosparc | doi:10.1038/nmeth.4169 | | |
| Software, algorithm | MotionCor2 | doi:10.1038/nmeth.4193 | | |
| Software, algorithm | ChimeraX | doi:10.1002/pro.3943 | | |
| Software, algorithm | CTFFIND4 | doi:10.1016/j.jsb.2015.08.008 | | |

## Cell culture

HEK293T cells were purchased from ATCC, routinely maintained in the lab without further authentication, and tested negative for mycoplasma. HEK293T cells endogenously expressing PAC channel, or PAC KO HEK293T cells, generated previously using CRISPR technology (*Yang et al., 2019*) were maintained in Dulbecco's modified Eagle's medium (DMEM) supplemented with 10% fetal bovine serum (FBS) and 1% penicillin/streptomycin (P/S) at 37 °C in humidified 95% $CO_2$ incubator. All PAC mutants mentioned in this manuscript were expressed and recorded in the PAC KO HEK2935 cell line. PAC KO cells were transfected with 500–800 ng/ml of plasmid DNA using Lipofectamine 2000 (Life Technologies according to the manufacturer's instructions. Cells were seeded on 12 mm diameter Poly-L-lysine Sigma-Aldrich) coated glass coverslips and were recorded within 24 hr after seeding/transfection.

## Constructs and mutagenesis

Human PAC isoform 2 coding sequence (NP_060722), previously subcloned into pIRES2-EGFP vector (Clontech) using XhoI and EcoRI restriction enzyme sites (*Yang et al., 2019*), was used for whole-cell patch-clamp recording experiments. Zebrafish PAC coding sequence (NP_001278691) was subcloned into pIRES2-EGFP vector (Clontech) using NheI and EcoRI restriction enzyme sites. Mutations were introduced using sense and antisense oligos with 15 base pairs of homology on each side of the mutated site. Site-directed mutagenesis was carried out using QuikChange II XL site-directed mutagenesis kit (Agilent Technologies) according to the manufacturer's instructions. All constructs were confirmed by sequencing the entire open reading frame using Sanger sequencing.

## Sequence alignments

PAC multiple protein sequence alignments were created using Clustal Omega software (EMBL-EBI). Protein sequences from the following vertebrate species were obtained from UniProt (ID): human PAC (Q9H813), rat PAC (Q66H28), mouse PAC (Q9D771), frog PAC (Q0V9Z3), zebrafish PAC (Q7SY31), bovine PAC (Q2KHV2), orangutan PAC (Q5RDP8), chicken PAC (E1C5B3), and green anole PAC (G1KFB8).

## Lipids and chemicals

All lipids used in this paper were ordered from Avanti Polar Lipids, and dissolved in water or DMSO, depending on the chain length, to make stock solutions. If not stated otherwise, lipids were added at 10 µM concentration directly to the extracellular solution. Please refer to the table for the list of all the lipids used in this paper. Poly-L-Lysine (PLL) (Sigma-Aldrich) was added to the intracellular solution at 100 µg/ml.

## Electrophysiology

Whole-cell patch-clamp experiments were performed using the extracellular recording solution (ECS) containing (in mM): 145 NaCl, 2 $MgCl_2$, 2 KCl, 1.5 $CaCl_2$, 10 HEPES, 10 glucose. The osmolarity of the ECS solution was 300–310 mOsm/kg and the pH was titrated to 7.3 using NaOH. Acidic extracellular solutions contained the same ionic composition, except 5 mM sodium citrate was used as a buffer instead of HEPES, and the pH was adjusted using citric acid. ECS solutions were applied 100–200 μm away from the recording cell, using a gravity perfusion system with a small tip. Recording patch pipettes, made of borosilicate glass (Sutter Instruments), were pulled with a Model P-1000 multi-step puller (Sutter Instruments). The patch pipettes had a resistance of 2–4 MΩ when filled with an intracellular solution (ICS) containing (in mM): 135 CsCl, 2 $CaCl_2$, 1 $MgCl_2$, 5 EGTA, 4 MgATP, 10 HEPES. The osmolarity of the ICS solution was 280–290 mOsm/kg and pH was titrated to 7.2 using CsOH. $I_{Cl, H}$ recordings were acquired using voltage ramp pulses from –100 to + 100 mV. The time interval between two ramp pulses was 2 or 5 s at a speed of 1 mV/ms and the holding potential was 0 mV. All recordings were performed with a MultiClamp 700B amplifier and 1550B digitizer (Molecular Devices) at room temperature. Signals were filtered at 2 kHz, digitized at 10 kHz, and the series resistance was compensated for at least 80% (*Yang et al., 2019*).

## Data analysis

Electrophysiology data were analyzed using Clampfit 10.7. Statistical analysis was performed using GraphPad Prism 9 software. Comparison between two groups was carried out using an unpaired two-tailed Student's *t* test unless stated otherwise. Multiple group comparisons were performed using ordinary one-way analysis of variance (ANOVA). The significance level was set at p<0.05. All numerical data are shown as mean ± SEM. For the time-constant experiments, the currents were fit using a one-phase decay equation: Y=(Y0 - Plateau)*exp(-K*X)+Plateau, where Y0 was the time-point of adding $PIP_2$ to the cells. For the $IC_{50}$ values, the normalized data was fitted to the following sigmoidal 4PL equation, where X is log (concentration): Span = Top - Bottom; Y=Bottom + (Top-Bottom)/ (1+10^(($LogIC_{50}$-X)*HillSlope)) (*Ruan et al., 2020*).

## Protein expression and purification

The pEGC-hPAC plasmid containing the human PAC gene, a Strep-tag II tag, a thrombin cleavage site, an eGFP, and an 8xHis tag, was used for expressing PAC protein in mammalian cells using BacMam system (*Goehring et al., 2014*, *Ruan et al., 2020*). The bacmid was produced by transforming the DH10Bac cells with pEGC-hPAC plasmid. Positive white clones were selected from a Luria Broth (LB) plate with kanamycin (50 μg/mL), tetracycline (10 μg/mL), gentamicin (7 μg/mL), Bluo-gal (100 μg/mL Bluo-gal), and IPTG (40 μg/mL). Bacmid DNA was purified from LB cultures of the white colonies using the alkaline lysis method. The bacmid was then transfected into adherent Sf9 cells grown in Sf-900 II media (Gibco) using Cellfectin II reagent by following the manufacturer's recommended protocol. After 5 days, the media of Sf9 cell culture was filtered and stored as the P1 virus. Subsequently, the P2 virus was made by infecting suspension Sf9 cells grown in Sf-900 II media with P1 virus at a 1:5000 ratio (v/v). After 5 days, the media containing P2 virus was harvested, filtered, and stored at 4 °C with 1% fetal bovine serum (FBS). Mammalian cells (tsA-201 cell line) grown in FreeStyle 293 media (Gibco) supplemented with 1% FBS was used for protein expression. When suspension cells reached 3.5x10^6 cells/ml density, 10% (v/v) P2 virus was added to tsA-201 cells, and cells were allowed to grow for 8–12 hr at 37 °C. To boost protein expression, 5 mM sodium butyrate was added to the cell culture, and cells were allowed to grow for another 60 hr at 30 °C. The mammalian cells expressing PAC were then spun down at 6000 rpm for 15 min, and the pellet is stored at –80 °C until protein purification.

The cell pellet was resuspended in ice-cold TBS buffer (20 mM Tris pH 8 and 150 mM NaCl) with a protease inhibitor cocktail (1 mM PMSF, 0.8 μM aprotinin, 2 μg/ml leupeptin, 2 mM pepstatin A) and lysed by sonication. The debris was removed by centrifugation at 4000 rpm for 10 min at 4 °C. The supernatant underwent ultracentrifugation at 40,000 rpm for 1 hr and the cell membrane was collected. The membrane was solubilized in TBS buffer with 1% glyco-diosgenin (GDN) detergent (Anatrace) and the protease inhibitor cocktail for 1 hr at 4 °C with gentle rotation. The sample was clarified by ultracentrifugation at 40,000 rpm for 1 hr. The supernatant was subjected to immobilized metal affinity chromatography (IMAC) with talon resin (Takara Bio USA). The bound protein was

washed with TBS buffer containing 0.02% GDN and 20 mM imidazole and eluted with TBS buffer containing 0.02% GDN and 250 mM imidazole. The PAC protein was then concentrated to 1 ml using a 100 kDa concentrator. The sample was then mixed with soybean lipid extract (Anatrace) and His-tag free membrane scaffold protein 1E3D1 (*Denisov et al., 2007*) at a 1:200:3 molar ratio. The GDN detergent was removed through three rounds of biobeads (Bio-Rad) incubation at 4 °C. To remove 'empty' nanodiscs, the sample was filtered to remove biobeads and incubated with talon resin at 4 °C for another 1 hr. The volume of the sample was expanded to 25 ml by adding TBS buffer such that the imidazole concentration was at 10 mM. The resin was washed with TBS buffer containing 10 mM imidazole, and the protein was eluted with TBS buffer containing 250 mM imidazole. PAC-nanodisc protein was then concentrated to 500 µL using an Amicon Ultra-15 concentrator (100 kDa cutoff). Thrombin (0.03 mg/ml) was added to cleave GFP from the PAC protein at 4 °C overnight. PAC-nanodisc was further purified by size-exclusion chromatography (SEC) using TBS buffer. The peak fractions were concentrated to 5 mg/ml before making cryo-EM grids.

## Cryo-EM grid preparation

Purified human PAC protein in nanodiscs was first mixed with 1 mM diC$_8$-PI(4,5)P$_2$ (Avanti) on ice for 1 hr. The pH of the protein sample was adjusted to 4.0 by adding an acidic acid buffer (1 M, pH 3.5) at a 1:20 ratio (v/v). We also added 0.5 mM fluorinated octyl maltoside (Anatrace) to improve sample quality. An FEI Vitrobot Mark III was used for plunge-freezing. Specifically, a 3 µl aliquot of the protein sample was applied to a glow-discharged Quantifoil holey carbon grid (Au 300 2/1 mesh) (Electron Microscopy Sciences), blotted for 2 s, vitrified in liquid ethane, and transferred to liquid nitrogen for storage. The temperature and humidity of the chamber was kept at 18 °C and 100% throughout the grid preparation.

## Cryo-EM data collection

The cryo-EM grids were initially screened in an FEI Talos Arctica transmission electron microscope equipped with a K2 summit camera. High-resolution data collection was facilitated by the Pacific Northwest Center for Cryo-EM (PNCC) using an FEI Titan Krios transmission electron microscope equipped with a BioQuantum energy filter (20 eV slit width) and a K3 camera with a nominal magnification of 105,000. SerialEM was used for automated data collection in super-resolution mode with a pixel size of 0.413 Å (*Mastronarde, 2005*). The raw movie stack contained a total of 52 frames with a total dose of 50 e$^-$/Å$^2$. The nominal defocus value was allowed to vary between –0.6 and –2.4 µ m.

## Cryo-EM data processing

The cryo-EM data processing workflow is summarized in *Figure 3—figure supplement 2*. Specifically, the raw movies were motion corrected using relion 3.1 and binned to the physical pixel size at 0.826 Å (*Zivanov et al., 2018*). The defocus parameters of motion-corrected micrographs were estimated using ctffind 4.1.10 (*Rohou and Grigorieff, 2015*). Particle picking was performed using both gautomatch_v0.56 and topaz v0.2.5 (*Bepler et al., 2019*). Particles picked by each program were independently subjected to 2D classification (relion 3.1) or heterogeneous refinement with C1 symmetry (cryosparc v3.0) to get rid of junk particles (*Punjani et al., 2017*; *Zivanov et al., 2018*). Good particles with clear features were pooled together and refined in relion 3.1. 3D refinement with a solvent mask and C3 symmetry, resulting in a 4.4 Å map. We noticed that the size of nanodiscs could be heterogeneous, which may negatively affect particle alignment. Therefore, we created a loose mask of the protein based on the atomic model and performed signal subtraction to remove the nanodisc signal. The process allowed us to obtain a reconstruction at 4.2 Å resolution. To sort out the conformational heterogeneity of the dataset, we performed 3D classification without image alignment in relion 3.1. The best class of the job was selected and refined to 3.6 Å resolution. We then performed several rounds of CTF refinement and Bayesian polishing (*Zivanov et al., 2019*), and the map was eventually refined to 3.17 Å. We noticed an improvement in the map quality when the box size of the images was expanded from 240 pixels to 300 pixels at this stage. To further improve map reconstruction, we first split the original consensus particles after 2D and heterogeneous refinement into 6 portions. We combined each portion with the best particles that gave rise to the 3.17 Å reconstruction and performed another round of 3D classification. This procedure was effective in attracting good particles from the initial consensus map particles. After combining the best class and removing

duplicates, we identified 84 k particles that could be refined to 3.07 Å in relion after iterative CTF refinement and Bayesian Polishing. We then exported the particles to cryosparc and conducted CTF refinement followed by local refinement. We also supplied a mask to get rid of nanodisc signal during the refinement. In the end, we obtained a 2.71 Å reconstruction as judged by gold-standard Fourier shell correlation. This map is deposited in the EMDB under the accession EMD-28535. We noticed that the intracellular side of TMD is very heterogeneous, which may limit the quality of the final reconstruction. As the last step of our data analysis, we refined the particles by using a mask that excludes signals of the intracellular region of PAC. This additional step allowed us to obtain a reconstruction of PAC ECD and the extracellular region of TMD at 2.70 Å (*Figure 3—figure supplement 2*). Although the nominal resolution of this map is comparable to the full protein map, we noticed improved map quality, especially the lipid density. Postprocessing of the maps, including local map sharpening and resolution estimation is performed using Phenix (*Terwilliger et al., 2018*). This map is deposited in the EMDB under the accession EMD-28964.

## Model building, validation, and analysis

The atomic model was generated by first docking the structural model of human PAC at pH 4.5 (PDBID: 7SQH) into the cryo-EM map (*Wang et al., 2022*). The diC$_8$-PI(4,5)P$_2$ molecule was manually placed into the cryo-EM density. The Grade Web Server (*Bepler et al., 2019*) was used to generate a restraint file for flexible fitting of the diC$_8$-PI(4,5)P$_2$ molecule. Subsequently, the model underwent real space refinement in phenix and manual adjustment to fix Ramachandran outliers, rotamer outliers, and clashes (*Adams et al., 2010*). We manually removed the phosphatidylinositol 4,5-bisphosphate group from the atomic model due to the limited support from the cryo-EM density. The final model was validated by the molprobity in phenix to obtain validation statistics (*Williams et al., 2018*). The cryo-EM map and atomic model were visualized using UCSF ChimeraX (*Pettersen et al., 2021*). The pore profile of the PAC channel was calculated using the HOLE 2.0 program (*Smart et al., 1996*).

## Acknowledgements

We thank the Qiu lab for thoughtful discussions. We thank G Zhao and X Meng for support with preliminary cryo-EM grid screening at the David Van Andel Advanced Cryo-Electron Microscopy Suite. L.M. is supported by a Boehringer Ingelheim Fonds (BIF) and National Institute of General Medical Sciences, T32 GM007445 (to the BCMB graduate training program). Z.R. is supported by an American Heart Association (AHA) postdoctoral fellowship (grant 20POST35120556) and the National Institute of Health (NIH) (grant K99NS128258). J O.-O. is supported by an AHA predoctoral fellowship (grant 18PRE34060025). W.L. is supported by the NIH (grant R01NS112363). Z.Q. is supported by a McKnight Scholar Award, a Klingenstein-Simon Scholar Award, a Sloan Research Fellowship in Neuroscience, and NIH grants (R35GM124824 and R01NS118014). A portion of this research was supported by NIH grant U24GM129547 and performed at the PNCC at OHSU and accessed through EMSL (grid.436923.9), a DOE Office of Science User Facility sponsored by the Office of Biological and Environmental Research.

## Additional information

### Funding

| Funder | Grant reference number | Author |
| --- | --- | --- |
| Boehringer Ingelheim Fonds | Graduate Student Fellowship | Ljubica Mihaljević |
| National Institute of General Medical Sciences | T32 GM007445 Graduate Training Program | Ljubica Mihaljević |
| American Heart Association | Postdoctoral Fellowship grant 20POST35120556 | Zheng Ruan |
| National Institutes of Health | grant K99NS128258 | Zheng Ruan |

| Funder | Grant reference number | Author |
|--------|------------------------|--------|
| American Heart Association | Predoctoral Fellowship grant 18PRE34060025 | James Osei-Owusu |
| National Institutes of Health | grant R01NS112363 | Wei Lü |
| McKnight Foundation | McKnight Scholar Award | Zhaozhu Qiu |
| Alfred P. Sloan Foundation | Sloan Research Fellowship | Zhaozhu Qiu |
| Esther A. and Joseph Klingenstein Fund | Klingenstein-Simons Fellowship | Zhaozhu Qiu |
| National Institutes of Health | grant R35GM124824 | Zhaozhu Qiu |
| National Institutes of Health | grant R01NS118014 | Zhaozhu Qiu |

The funders had no role in study design, data collection and interpretation, or the decision to submit the work for publication.

### Author contributions

Ljubica Mihaljević, Conceptualization, Data curation, Formal analysis, Validation, Investigation, Visualization, Methodology, Writing – original draft, Writing – review and editing; Zheng Ruan, Conceptualization, Data curation, Software, Formal analysis, Validation, Investigation, Visualization, Methodology, Writing – review and editing; James Osei-Owusu, Data curation, Formal analysis, Validation, Investigation, Visualization, Methodology, Writing – review and editing; Wei Lü, Zhaozhu Qiu, Conceptualization, Resources, Supervision, Funding acquisition, Project administration, Writing – review and editing

### Author ORCIDs

Ljubica Mihaljević http://orcid.org/0000-0002-3697-7767
Zheng Ruan http://orcid.org/0000-0002-4412-4916
Wei Lü http://orcid.org/0000-0002-3009-1025
Zhaozhu Qiu http://orcid.org/0000-0002-9122-6077

### Decision letter and Author response

Decision letter https://doi.org/10.7554/eLife.83935.sa1
Author response https://doi.org/10.7554/eLife.83935.sa2

## Additional files

### Supplementary files

• Supplementary file 1. Cryo-EM data collection, refinement, and validation statistics.
• MDAR checklist

### Data availability

The cryo-EM density maps have been deposited in the EMDB (Electron Microscopy Data Bank) under accession numbers EMD-28535 and EMD-28964. The atomic models have been deposited in the Research Collaboratory for Structural Bioinformatics Protein Data Bank (RCS-PDB) under accession numbers 8EQ4 and 8FBL.

The following datasets were generated:

| Author(s) | Year | Dataset title | Dataset URL | Database and Identifier |
|-----------|------|---------------|-------------|-------------------------|
| Mihaljevic L, Ruan Z, Osei-Owusu J, Lu W, Qiu Z | 2022 | Cryo-EM structure of PAC channel with PIP2 | https://www.ebi.ac.uk/emdb/EMD-28535 | Electron Microscopy Data Bank, EMD-28535 |

*Continued on next page*

*Continued*

| Author(s) | Year | Dataset title | Dataset URL | Database and Identifier |
|---|---|---|---|---|
| Mihaljevic L, Ruan Z, Osei-Owusu J, Lu W, Qiu Z | 2022 | Cryo-EM structure of PAC channel with PIP2 | https://www.ebi.ac.uk/emdb/EMD-28964 | Electron Microscopy Data Bank, EMD-28964 |
| Mihaljevic L, Ruan Z, Osei-Owusu J, Lu W, Qiu Z | 2022 | Cryo-EM structure of PAC channel with PIP2 | https://www.rcsb.org/structure/8EQ4 | RCSB Protein Data Bank, 8EQ4 |
| Mihaljevic L, Ruan Z, Osei-Owusu J, Lu W, Qiu Z | 2022 | Cryo-EM structure of PAC channel with PIP2 | https://www.rcsb.org/structure/8FBL | RCSB Protein Data Bank, 8FBL |

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
