## [Editor Report]

The recently identified Proton-Activated Chloride (PAC) channel is ubiquitously expressed and has important roles in intracellular organelles and its function in the plasma membrane is associated with human pathologies. Combining electrophysiology, site-directed mutagenesis, lipid pharmacology, and single particle cryo-electron microscopy, this valuable study provides solid evidence to identify a site on the extracellular half of the transmembrane domain of PAC channels that could be occupied by PIP2 and related lipids to promote channel desensitization. These findings are relevant because pharmacological information for this important ion channel is absent.

---

## [Decision Letter]

**Decision letter after peer review:**

Thank you for submitting your article "Inhibition of the proton-activated chloride channel PAC by PIP2" for consideration by *eLife*. Your article has been reviewed by 3 peer reviewers, including Andrés Jara-Oseguera as Reviewing Editor and Reviewer #1, and the evaluation has been overseen by Richard Aldrich as the Senior Editor. The following individual involved in the review of your submission has agreed to reveal their identity: Valeria Kalienkova (Reviewer #2).

The reviewers have discussed their reviews with one another, and the Reviewing Editor has drafted this to help you prepare a revised submission. All three reviewers agreed that the findings in the manuscript are interesting and important, but they also raised a series of concerns, the two major ones concerning whether the inhibition of PAC channels by PIP2 is physiologically relevant, and whether the lipid bound to the channel in the structure is indeed PIP2. Each of the concerns discussed by the reviewers are summarized below.

Essential revisions:

1. Since the authors show that PIP2 rather acts on an activated/desensitized state of the channel, the authors should discuss the state of the PIP2 binding site in different states of the channel – resting, activated and desensitized. Is it unavailable to PIP2 in resting state? Related to this, do previously solved desensitized structures have any defined lipid densities in that region? If this is not the case, this might be worth mentioning to further strengthen the point that the density observed in this new structure might indeed be a specifically-bound PIP2 molecule.

2. For Figure 3C (PIP2 binding site), please include the cryo-EM density for the lipid, the fitted PIP2 molecule, and the density for the side-chains that are interacting with the lipid. The experimental evidence identifying PIP2 as the bound lipid should be discussed more extensively, and it should be clearly acknowledged in the paper if there are alternative explanations for the observations: does the density uniquely identify the lipid as PIP2, or could other lipids be fitted, including potential lipid contaminants such as phosphatidic acid? The authors need to clarify whether their experimentally resolved data supports the positioning of bound PIP2 in the PAC structure (Figure 3A-C) or does not (the main text and figure legends, Figure S4).

Please also clarify if the longer acyl chains are proposed to contribute to tighter binding of the lipid and orientation of the headgroup into the pocket. This is important because the headgroup does not seem to bind very tightly and is hence not very well resolved. Is the pocket big enough to accommodate PIP3? Does the higher number of phosphates simply help to attract the headgroup towards this pocket? Finally, note the pKA of inositol phosphates (Kooijman et al. 2009, PMID: 19725516); they will be partially protonated at pH 4.

3. The authors need to address the subcellular localization of PI(4,5)P2 and why their findings are surprising. Although they cite previous studies that have identified PI(4,5)P2 on the outer leaflet of the plasma membrane, it is a minor component of total PIP2 (see Yoneda et al. 2020, Figure 1). The localization and function of cellular PIP2 has been extensively studied (reviewed in Schink et al. 2016, PMID 27576122) and few roles for extracellular/lumenal PIP2 have been described to date. It is unlikely (albeit possible) that a native regulatory mechanism for PIP2 exists in this context, and the conclusions of the paper should reflect this fact.

4. While the unsharpened map looks symmetric, the sharpened one does not – what is the reason for this discrepancy? Please indicate in Materials and methods if C3 symmetry was applied throughout the refinement for the 4.2 A map. The authors might consider trying other sharpening tools which take into account different b-factors across the map (deepEMhancer, local deblur, sharpening tools in Phenix etc). Please include the reasoning for the final mask excluding part of the transport domain in data processing section of Materials and methods, and display the mask in the processing workflow in figure S3. Regarding figure S3, please indicate what is displayed in color, and what are the transparent outlines – are those depictions of the maps at different contour? Or are the transparent outlines masks used for sharpening? If it is the latter, the final mask also includes nanodisc density.

5. Application of PIP2 in Figure 4A-C produces an immediate drop of current that appears independent from the decay curves ascribed to desensitization. The brackets on those figures indicate that inhibition was measured 75-100 seconds after application, thus incorporating a mix of both processes. This measurement needs to be consistent across experiments and should be explained more clearly. Also, the rate of desensitization before and after PIP2 does not seem to change after the rapid current decay upon first exposure to PIP2 – this seems inconsistent with the proposed mechanism of inhibition by PIP2 in which the lipid increases the rate of desensitization. Please show the fits for the one-phase decay equations. The drop of current visible after PIP2 application suggests that such a model is inappropriate. These discrepancies need to be appropriately addressed. In the discussion, "PIP2 could only slowly shift the equilibrium toward the desensitized state" is not supported by the data in Figure 4.

6. Experiments with the PAC KO cell line should be included showing that the increase in current caused by low pH, and its inhibition by applied PIP2 are only observed in cells that were transfected with WT PAC channels.

7. It must be clarified whether the PAC KO cell line was used for all experiments with mutants.

8. Table S1, map resolution range: is 246 A a typo? This seems unusually low.

9. Although not necessary, it would strengthen the manuscript if measurements of the desensitization rate for key mutants in Figure 3E were included.

10. In the introduction, a citation is needed for the statement "TMEM16A is known to require PIP2 for channel opening."

11. No significant difference is reported between the IC50 of PIP2 and PIP3. The text "the additional phosphate on PIP3 further lowered the IC50…" is not supported by the data.

12. In Figure 2(D), diC18:1-PI(4,5)P2 is incorrectly described as natural.

13. Also in Figure 2(D), poor solubility is used to justify using diC18:1-PI(4,5)P2 at higher concentration. This doesn't make sense; higher concentrations reduce solubility. The authors should consider addressing this discrepancy.

14. In the discussion, "the desensitized conformation becomes more prevalent and accessible" should be changed to reflect that the binding site becomes more accessible.

15. In the discussion, ABCA1 is proposed as a shuttle for PIP2 in cells expressing PAC channels. Do these proteins co-express?

16. The manuscript states that PAC/ASOR is the first Cl^-^ channel to be inhibited by PIP2. TMEM16B is inhibited by PIP2, reported first in the following manuscript: Ta et al. 2017, PMID 28616863.

17. Please review the methods section for grammar and capitalization errors.

*Reviewer #1 (Recommendations for the authors):*

1) Experiments with the PAC KO cell line should be included showing that the increase in current caused by low pH, and its inhibition by applied PIP2 are only observed in cells that were transfected with WT PAC channels.

2) It must be clarified whether the PAC KO cell line was used for all experiments with mutants.

3) Including measurements of the desensitization rate for key mutants included in Figure 3E would strengthen the conclusions for a direct interaction between PIP2 and the proposed site on the channel.

4) A more critical discussion should be included that examines what other types of lipids could be occupying the density that is assigned to PIP2 in the study.

*Reviewer #2 (Recommendations for the authors):*

1. Since the authors show that PIP2 rather acts on an activated/desensitized state of the channel, it would be interesting to show the PIP2 binding site in different states of the channel – resting, activated and desensitized. Is it unavailable to PIP2 in resting state? Related to this, do previously solved desensitized structures have any defined lipid densities in that region? If this is not the case, this might be worth mentioning to further strengthen the point that the density observed in this new structure might indeed be a specifically-bound PIP2 molecule.

2. Description of the PIP2-binding site, page 7 bottom line: this could benefit from figure with a close-up of PIP2 binding site, with the corresponding cryo-EM density, and the fitted PIP2 molecule. Page 8, end of the first paragraph, I feel the explanation could be expanded further: do the longer acyl chains help tighter binding of the lipid and orientation of the headgroup into the pocket? Is this important because the headgroup does not bind very tightly and is hence not very well resolved? Is the pocket big enough to accommodate PIP3? Does the higher number of phosphates simply help to attract the headgroup towards this pocket?

3. While the unsharpened map looks symmetric, the sharpened one does not, what is the reason for this discrepancy? Perhaps also worth indicating in Materials and methods if from the point of 3D refinement which gave the 4.2 A map C3 symmetry was applied throughout. The authors might consider trying other sharpening tools which take into account different b-factors across the map (deepEMhancer, local deblur, sharpening tools in Phenix etc). It would also be valuable to include the reasoning for the final mask excluding part of the transport domain in data processing section of Materials and methods, and to display the mask in the processing workflow in figure S3. Regarding figure S3, please indicate what is displayed in color, and what are the transparent outlines – are those depictions of the maps at different contour? Or are the transparent outlines masks used for sharpening? If it is the latter, the final mask also includes nanodisc density, contrary to

4. While not absolutely necessary, it could be helpful to mention for those unfamiliar with this protein family that it is also referred to as ASOR.

5. Table S1, map resolution range: is 246 A a typo? This seems unusually low.

*Reviewer #3 (Recommendations for the authors):*

1) The authors need to address the subcellular localization of PI(4,5)P2 and why their findings are surprising. Although they cite previous studies that have identified PI(4,5)P2 on the outer leaflet of the plasma membrane, it is a minor component of total PIP2 (see Yoneda et al. 2020, Figure 1). The localization and function of cellular PIP2 has been extensively studied (reviewed in Schink et al. 2016, PMID 27576122) and few roles for extracellular/lumenal PIP2 have been described to date. It is unlikely (albeit possible) that a native regulatory mechanism for PIP2 exists in this context, and the conclusions of the paper should reflect this fact.

2) The density of PAC with PIP2 (Figure S4) does not support the placement of the PIP2 headgroup. The authors acknowledge this fact with the following points in the Results section: "it is by no means unambiguous due to the limited local map resolution" and "the phosphatidyl group is reasonably well defined… in contrast, the inositol 4,5-bisphosphate moiety is not resolved". Confusingly, the authors also state that the deposited model does not contain the inositol headgroup (Figure 3A) but the attached validation files indicate that it has been (Molecule 3, PIO "Ligand of interest"). The authors need to decide whether their experimentally resolved data supports the positioning of bound PIP2 in the PAC structure (Figure 3A-C) or does not (the main text and figure legends, Figure S4). In doing so, the authors should consider the possibility of contaminating ligands from the other lipids (e.g. phosphatidic acid) in their preparation. Furthermore, in absence of supporting data, Figure 3C should be excluded from the manuscript. While it is apparent that the modeled basic residues are important for PAC inhibition by PIP2 (Figure 3E) and the authors are careful to note that the model is used to highlight the local biochemical (electrostatic) environment, the presentation of the figure (e.g. distance lines for electrostatic interactions) suggests more certainty than is warranted. Finally, note the pKA of inositol phosphates (Kooijman et al. 2009, PMID: 19725516); they will be partially protonated at pH 4.

---

## [Author Response]

Essential revisions:1. Since the authors show that PIP2 rather acts on an activated/desensitized state of the channel, the authors should discuss the state of the PIP2 binding site in different states of the channel – resting, activated and desensitized. Is it unavailable to PIP2 in resting state? Related to this, do previously solved desensitized structures have any defined lipid densities in that region? If this is not the case, this might be worth mentioning to further strengthen the point that the density observed in this new structure might indeed be a specifically-bound PIP2 molecule.

The residues that form the PIP2 binding site are part of the structural region that shows major conformational differences between the desensitized state and the resting/activated states (Figure S5). In particular, the rotational movement of TM1 annihilates the PIP2 binding pocket in the resting and activated states of PAC (Figure S5). Therefore, it’s unlikely that PIP2 binds to PAC in these states. We thank the reviewer for raising such a good point and we now generate a new figure to further demonstrate this point (Figure S5).

Our previous structure in the desensitized state doesn't contain such a lipid density (Ruan et al., *Nature*, 2020). The recent high-resolution structure of PAC appears to contain a branched lipid density in the area in two of the subunits but not the third one. We are unsure about the reason as the map is supposed to be refined using C3 symmetry. The map deposited in the EMDB (EMD-25385) looks distorted and doesn’t appear to be a raw cryo-EM map generated directly by gold-standard 3D refinement. We are unsure how to best interpret the map (Wang et al., *Sci Adv*, 2022). Therefore, we decide not to discuss this part in our manuscript.

2. For Figure 3C (PIP2 binding site), please include the cryo-EM density for the lipid, the fitted PIP2 molecule, and the density for the side-chains that are interacting with the lipid. The experimental evidence identifying PIP2 as the bound lipid should be discussed more extensively, and it should be clearly acknowledged in the paper if there are alternative explanations for the observations: does the density uniquely identify the lipid as PIP2, or could other lipids be fitted, including potential lipid contaminants such as phosphatidic acid? The authors need to clarify whether their experimentally resolved data supports the positioning of bound PIP2 in the PAC structure (Figure 3A-C) or does not (the main text and figure legends, Figure S4).Please also clarify if the longer acyl chains are proposed to contribute to tighter binding of the lipid and orientation of the headgroup into the pocket. This is important because the headgroup does not seem to bind very tightly and is hence not very well resolved. Is the pocket big enough to accommodate PIP3? Does the higher number of phosphates simply help to attract the headgroup towards this pocket? Finally, note the pKA of inositol phosphates (Kooijman et al. 2009, PMID: 19725516); they will be partially protonated at pH 4.

We have revised Figure 3C to show the density of the lipid and important residues nearby. We also mentioned the possibilities of other lipids in the text. In our initial manuscript, we already acknowledged the fact that the assignment of PIP2 into the density is not unambiguous based on the cryo-EM map alone. However, the biochemical environment of the binding pocket, as well as the subsequent mutagenesis data support such a PIP2 binding site. We now re-wrote the paragraph to make this even clearer.

We propose that lipid chain of at least 8 carbons is necessary for PAC inhibition due to insertion into the membrane, which potentially makes the PIP2 headgroup accessible to the positively charged pocket on PAC. Lipid with a shorter chain, diC6-PIP2, didn’t inhibit PAC and the lipid with a chain longer than 8 carbons, diC18:0-20:4-PIP2, didn’t further increase the inhibition (when compared to diC8-PIP2), suggesting that C7 and C8 carbon of the acyl chain establish critical interactions with the protein. Additionally, our structural model showed that the acyl chain of PIP2 tightly packs against W304. The importance of the acyl chain for PIP2 binding is supported by the fact that the W304A mutant resulted in reduced PIP2 inhibition (Figure 3E).

The binding pocket is big enough to accommodate PIP3. In fact, we can place a phosphate group at the 3rd position of the inositol hydroxyl group in our current PDB model without creating any steric hindrance. Because the electrostatic environment surrounding the headgroup of PIP2 is positively charged (Figure 3C), we believe the attraction effect is a major factor that contributes to the recognition of PIP2.

We acknowledge the referenced literature (Kooijman et al. 2009, PMID: 19725516) that the phosphate group may be partially protonated at pH 4, but this study is unable to reliably estimate the pK_a2_ of PI(4,5)P2 and the pK_a1_ is not studied. Moreover, in Figure 7 of the paper, no major difference is observed in the charge status of PI(4,5)P2 at pH 4-5, suggesting that the ionization status of PIP_2_ doesn’t change in the experimental condition investigated in our study. In a more recent review of PI(4,5)P2, it was suggested that at pH 4-5, both of the phosphomonoester groups are mono-protonated (Luís Borges-Araújo and Fabio Fernandes 2020, PMID: 32858905). Therefore, a net charge of -2 is present in the head group of PI(4,5)P2 under our experimental conditions, which allowed the molecule to interact favorably with the positively charged binding pocket (Figure 3C).

3. The authors need to address the subcellular localization of PI(4,5)P2 and why their findings are surprising. Although they cite previous studies that have identified PI(4,5)P2 on the outer leaflet of the plasma membrane, it is a minor component of total PIP2 (see Yoneda et al. 2020, Figure 1). The localization and function of cellular PIP2 has been extensively studied (reviewed in Schink et al. 2016, PMID 27576122) and few roles for extracellular/lumenal PIP2 have been described to date. It is unlikely (albeit possible) that a native regulatory mechanism for PIP2 exists in this context, and the conclusions of the paper should reflect this fact.

Thank you for this insight, we agree that PIP2 predominantly localizes to the inner leaflets of the cell membrane. We have further clarified in the Discussion section of our paper that the physiological relevance (if any exists) of PIP2 inhibition of the PAC channel is unclear. Nevertheless, the novel lipid-binding pocket is a relevant discovery that can be exploited for targeted inhibition of the PAC channel.

4. While the unsharpened map looks symmetric, the sharpened one does not – what is the reason for this discrepancy? Please indicate in Materials and methods if C3 symmetry was applied throughout the refinement for the 4.2 A map. The authors might consider trying other sharpening tools which take into account different b-factors across the map (deepEMhancer, local deblur, sharpening tools in Phenix etc). Please include the reasoning for the final mask excluding part of the transport domain in data processing section of Materials and methods, and display the mask in the processing workflow in figure S3. Regarding figure S3, please indicate what is displayed in color, and what are the transparent outlines – are those depictions of the maps at different contour? Or are the transparent outlines masks used for sharpening? If it is the latter, the final mask also includes nanodisc density.

We thank the reviewer for pointing this out. We now carefully performed local anisotropic sharpening in Phenix and ensured that the output is symmetric. We also tried the deepEMhancer and local deblur. Upon visual inspection of the output, we found that the map produced by Phenix showed the best features in terms of the residue side chain and lipid density. Therefore, we deposit the Phenix map in EMDB.

We did apply C3 symmetry to obtain the initial 4.2 Å map. This is now clearly indicated in the Materials and methods.

Our final data processing procedure includes a step to exclude part of the map by using a mask. This is because substantial heterogeneity is present in the intracellular side of the transmembrane domain,. As we are primarily interested in the PIP2 binding site, which is located in the ECD-TMD interface, we decide to exclude signals in the intracellular region of the transmembrane domain for final refinement. This will allow us to improve the density for the PIP2 binding site and facilitate model building. We deposited this reconstruction as well as the full PAC map without using such a mask in EMDB in this revision.

The transparent outline is a non-sharpened map at a low contour level with the aim to show the nanodisc density. This will allow the readers to distinguish different regions of the PAC structure (ECD and TMD) easily. It is not a mask for refinement. We modified the figure S3 legend to better describe this. As the reviewer suggested, we also showed the mask used in the last step of refinement in Figure S3. This mask excluded the intracellular portion of PAC TMD.

5. Application of PIP2 in Figure 4A-C produces an immediate drop of current that appears independent from the decay curves ascribed to desensitization. The brackets on those figures indicate that inhibition was measured 75-100 seconds after application, thus incorporating a mix of both processes. This measurement needs to be consistent across experiments and should be explained more clearly. Also, the rate of desensitization before and after PIP2 does not seem to change after the rapid current decay upon first exposure to PIP2 – this seems inconsistent with the proposed mechanism of inhibition by PIP2 in which the lipid increases the rate of desensitization. Please show the fits for the one-phase decay equations. The drop of current visible after PIP2 application suggests that such a model is inappropriate. These discrepancies need to be appropriately addressed. In the discussion, "PIP2 could only slowly shift the equilibrium toward the desensitized state" is not supported by the data in Figure 4.

The reviewer raises a good point. Although the immediate drop in current is a consequence of recording resolution (5s/sweep), we have concluded that using one or two-phase decay equations are indeed not appropriate models to fit the PAC inhibition kinetics by PIP2. We, therefore, decided to exclude the kinetics from the manuscript and revised our proposed model appropriately. Furthermore, we wanted to clarify that the inhibition has been consistently quantified at 100s point after the application of PIP2, even though the start of PIP2 application had a variable time-point due to variability in desensitization current reaching a plateau. This is now clarified in the text.

6. Experiments with the PAC KO cell line should be included showing that the increase in current caused by low pH, and its inhibition by applied PIP2 are only observed in cells that were transfected with WT PAC channels.

We are thankful to the reviewer for pointing this out, however, we and others have shown repeatedly (Yang et al., 2019; PMID: 31023925, Osei-Owusu et al., 2022 PMID: 35878032; Osei-Owusu et al., 2021 PMID: 33503418; Ullrich et al., 2019; PMID: 31318332) that the chloride current elicited by low pH is absent in the PAC KO cells. In other words, there is no acid-induced chloride current (hence no current to be inhibited by PIP2) to be observed in PAC KO cells. Therefore, the currents recorded in this manuscript are PAC-specific and we believe that additional experiments are not necessary.

7. It must be clarified whether the PAC KO cell line was used for all experiments with mutants.

Thank you for this comment, we clarified further in the Materials and methods of our manuscript that all PAC mutants in the manuscript have been expressed and recorded in the PAC KO HEK293T cell line we reported previously.

8. Table S1, map resolution range: is 246 A a typo? This seems unusually low.

We thank the reviewers for pointing this out. This is indeed a typo and we have fixed this in Table S1.

9. Although not necessary, it would strengthen the manuscript if measurements of the desensitization rate for key mutants in Figure 3E were included.

Reviewer raised a good point. We quantified the amount of desensitization between the WT and the mutants at pH 5.0 (which showed no significant difference) and included it in the supplementary Figure 2F.

10. In the introduction, a citation is needed for the statement "TMEM16A is known to require PIP2 for channel opening."

We are thankful to the reviewer for this remark, we changed this sentence to include the new literature and added the appropriate citations.

11. No significant difference is reported between the IC50 of PIP2 and PIP3. The text "the additional phosphate on PIP3 further lowered the IC50…" is not supported by the data.

Thank you for pointing this out, we have made the following change: "The additional phosphate on PIP3 yielded the IC50 of 3μM " as supported by the data in the Figure 2B.

12. In Figure 2(D), diC18:1-PI(4,5)P2 is incorrectly described as natural.

Thank you for raising this point. We used a full-length diC18:0-20:0-PI(4,5)P2 instead.

13. Also in Figure 2(D), poor solubility is used to justify using diC18:1-PI(4,5)P2 at higher concentration. This doesn't make sense; higher concentrations reduce solubility. The authors should consider addressing this discrepancy.

We agree with the reviewer and we have addressed this discrepancy by performing a new experiment with 10M diC18:0-20:0-PI(4,5)P2. We have found that at this concentration full-length PIP2 potently inhibits the PAC channel.

14. In the discussion, "the desensitized conformation becomes more prevalent and accessible" should be changed to reflect that the binding site becomes more accessible.

Thank you for this comment, we have made the appropriate change in the discussion to reflect this.

15. In the discussion, ABCA1 is proposed as a shuttle for PIP2 in cells expressing PAC channels. Do these proteins co-express?

We are thankful to the reviewer for raising this point. PAC shows a broad expression across different tissues (Yang et al., 2019; PMID: 31023925; Ullrich et al., 2019; PMID: 31318332) and so does ABCA1 (Fagerberg et al., 2014; PMID 24309898), therefore the coincidence of co-expression between these two proteins is high.

16. The manuscript states that PAC/ASOR is the first Cl^-^ channel to be inhibited by PIP2. TMEM16B is inhibited by PIP2, reported first in the following manuscript: Ta et al. 2017, PMID 28616863.

We are thankful to the reviewer for this insight, and we have made the changes in our manuscript to reflect this literature.

17. Please review the methods section for grammar and capitalization errors.

We have addressed this.

Reviewer #1 (Recommendations for the authors):1) Experiments with the PAC KO cell line should be included showing that the increase in current caused by low pH, and its inhibition by applied PIP2 are only observed in cells that were transfected with WT PAC channels.

We are thankful to the reviewer for pointing this out, however, we and others have shown repeatedly (Yang et al., 2019; PMID: 31023925, Osei-Owusu et al., 2022 PMID: 35878032; Osei-Owusu et al., 2021 PMID: 33503418; Ullrich et al., 2019; PMID: 31318332) that the current elicited by low pH is absent in the PAC KO cells. In other words, there is no acid-induced chloride current (hence no current to be inhibited by PIP2) to be observed in PAC KO cells. Therefore, the currents recorded in this manuscript are PAC-specific and we believe that additional experiments are not necessary.

2) It must be clarified whether the PAC KO cell line was used for all experiments with mutants.

Thank you for this comment, we clarified further in the Materials and methods of our manuscript that all PAC mutants in the manuscript have been expressed and recorded in the PAC KO HEK293T cell line we reported previously.

3) Including measurements of the desensitization rate for key mutants included in Figure 3E would strengthen the conclusions for a direct interaction between PIP2 and the proposed site on the channel.

Reviewer raised a good point. We quantified the amount of desensitization between the WT and the mutants at pH 5.0 (which showed no significant difference) and included it in the supplementary Figure 2F.

4) A more critical discussion should be included that examines what other types of lipids could be occupying the density that is assigned to PIP2 in the study.

We are grateful to the reviewer for making this point. We of course cannot completely exclude the possibility that this density may represent other types of lipids, such as phosphatidic acid. We have now specifically mentioned this in the text.

Reviewer #2 (Recommendations for the authors):1. Since the authors show that PIP2 rather acts on an activated/desensitized state of the channel, it would be interesting to show the PIP2 binding site in different states of the channel – resting, activated and desensitized. Is it unavailable to PIP2 in resting state? Related to this, do previously solved desensitized structures have any defined lipid densities in that region? If this is not the case, this might be worth mentioning to further strengthen the point that the density observed in this new structure might indeed be a specifically-bound PIP2 molecule.

The residues that form the PIP2 binding site are part of the structural region that shows major conformational differences between the desensitized state and the resting/activated states (Figure S5). In particular, the rotational movement of TM1 annihilates the PIP2 binding pocket in the resting and activated states of PAC (Figure S5). Therefore, it’s unlikely that PIP2 binds to PAC in these states. We thank the reviewer for raising such a good point and we now generate a new figure to further demonstrate this point (Figure S5).

Our previous structure in the desensitized state doesn't contain such a lipid density (Ruan et al., *Nature*, 2020). The recent high-resolution structure of PAC appears to contain a branched lipid density in the area in two of the subunits but not the third one. We are unsure about the reason as the map is supposed to be refined using C3 symmetry. The map deposited in the EMDB (EMD-25385) looks distorted and doesn’t appear to be a raw cryo-EM map generated directly by gold-standard 3D refinement. We are unsure how to best interpret the map (Wang et al., *Sci Adv*, 2022). Therefore, we decide not to discuss this part in our manuscript.

2. Description of the PIP2-binding site, page 7 bottom line: this could benefit from figure with a close-up of PIP2 binding site, with the corresponding cryo-EM density, and the fitted PIP2 molecule. Page 8, end of the first paragraph, I feel the explanation could be expanded further: do the longer acyl chains help tighter binding of the lipid and orientation of the headgroup into the pocket? Is this important because the headgroup does not bind very tightly and is hence not very well resolved? Is the pocket big enough to accommodate PIP3? Does the higher number of phosphates simply help to attract the headgroup towards this pocket?

We have revised Figure 3C to show the density of the lipid and important residues nearby. In our initial manuscript, we already acknowledged the fact that the assignment of PIP2 into the density is not unambiguous based on the cryo-EM map. However, the biochemical environment of the binding pocket, as well as the subsequent mutagenesis data support such a PIP2 binding site. We now re-wrote the paragraph to make this even clearer.

We propose that lipid chain of at least 8 carbons is necessary for PAC inhibition due to insertion into the membrane, which potentially makes the PIP2 headgroup accessible to the positively charged pocket on PAC. Lipid with a shorter chain, diC6-PIP2, didn’t inhibit PAC and the lipid with a chain longer than 8 carbons, diC18:0-20:4-PIP2, didn’t further increase the inhibition (when compared to diC8-PIP2), suggesting that C7 and C8 carbon of the acyl chain establish critical interactions with the protein. Additionally, our structural model showed that the acyl chain of PIP2 tightly packs again W304. The importance of the acyl chain for PIP2 binding is supported by the fact that the W304A mutant resulted in reduced PIP2 inhibition.

The binding pocket is big enough to accommodate PIP3. In fact, we can place a phosphate group at the 3rd position of the inositol hydroxyl group in our current PDB model without creating any steric hindrance. Because the electrostatic environment surrounding the headgroup of PIP2 is positively charged (Figure 3C), we believe the attraction effect is a major factor that contributes to the recognition of PIP2.

3. While the unsharpened map looks symmetric, the sharpened one does not, what is the reason for this discrepancy? Perhaps also worth indicating in Materials and methods if from the point of 3D refinement which gave the 4.2 A map C3 symmetry was applied throughout. The authors might consider trying other sharpening tools which take into account different b-factors across the map (deepEMhancer, local deblur, sharpening tools in Phenix etc). It would also be valuable to include the reasoning for the final mask excluding part of the transport domain in data processing section of Materials and methods, and to display the mask in the processing workflow in figure S3. Regarding figure S3, please indicate what is displayed in color, and what are the transparent outlines – are those depictions of the maps at different contour? Or are the transparent outlines masks used for sharpening? If it is the latter, the final mask also includes nanodisc density, contrary to

We thank the reviewer for pointing this out. We now carefully performed local anisotropic sharpening in Phenix and ensured that the output is symmetric. We also tried the deepEMhancer and local deblur. Upon visual inspection of the output, we found that the map produced by Phenix showed the best features in terms of the residue side chain and lipid density. Therefore, we deposit the Phenix map in EMDB.

We did apply C3 symmetry to obtain the initial 4.2 Å map. This is now clearly indicated in the Materials and methods.

Our final data processing procedure includes a step to exclude part of the map by using a mask. This is because substantial heterogeneity is present in the intracellular side of the transmembrane domain,. As we are primarily interested in the PIP2 binding site, which is located in the ECD-TMD interface., we decide to exclude signals in the intracellular region of the transmembrane domain for final refinement. This will allow us to improve the density for the PIP2 binding site and facilitate model building. We deposited this reconstruction as well as the full PAC map without using such a mask in EMDB in this revision.

The transparent outline is a non-sharpened map at a low contour level with the aim to show the nanodisc density. This will allow the readers to distinguish different regions of the PAC structure (ECD and TMD) easily. It is not a mask for refinement. We modified the figure S3 legend to better describe this. As the reviewer suggested, we also showed the mask used in the last step of refinement in Figure S3. This mask excluded the intracellular portion of PAC TMD.

4. While not absolutely necessary, it could be helpful to mention for those unfamiliar with this protein family that it is also referred to as ASOR.

Thank you for pointing this out, we have mentioned that this protein family is also referred to as ASOR in the introduction.

5. Table S1, map resolution range: is 246 A a typo? This seems unusually low.

We thank the reviewers for pointing this out. This is indeed a typo and we have fixed this in Table S1.

Reviewer #3 (Recommendations for the authors):1) The authors need to address the subcellular localization of PI(4,5)P2 and why their findings are surprising. Although they cite previous studies that have identified PI(4,5)P2 on the outer leaflet of the plasma membrane, it is a minor component of total PIP2 (see Yoneda et al. 2020, Figure 1). The localization and function of cellular PIP2 has been extensively studied (reviewed in Schink et al. 2016, PMID 27576122) and few roles for extracellular/lumenal PIP2 have been described to date. It is unlikely (albeit possible) that a native regulatory mechanism for PIP2 exists in this context, and the conclusions of the paper should reflect this fact.

Thank you for this insight, we agree that PIP2 predominantly localizes to the inner leaflets of the cell membrane. We have further clarified in the Discussion section of our paper that the physiological relevance (if any exists) of PIP2 inhibition of the PAC channel is unclear. Nevertheless, the novel lipid-binding pocket is a relevant discovery that can be exploited for targeted inhibition of the PAC channel.

2) The density of PAC with PIP2 (Figure S4) does not support the placement of the PIP2 headgroup. The authors acknowledge this fact with the following points in the Results section: "it is by no means unambiguous due to the limited local map resolution" and "the phosphatidyl group is reasonably well defined… in contrast, the inositol 4,5-bisphosphate moiety is not resolved". Confusingly, the authors also state that the deposited model does not contain the inositol headgroup (Figure 3A) but the attached validation files indicate that it has been (Molecule 3, PIO "Ligand of interest"). The authors need to decide whether their experimentally resolved data supports the positioning of bound PIP2 in the PAC structure (Figure 3A-C) or does not (the main text and figure legends, Figure S4). In doing so, the authors should consider the possibility of contaminating ligands from the other lipids (e.g. phosphatidic acid) in their preparation. Furthermore, in absence of supporting data, Figure 3C should be excluded from the manuscript. While it is apparent that the modeled basic residues are important for PAC inhibition by PIP2 (Figure 3E) and the authors are careful to note that the model is used to highlight the local biochemical (electrostatic) environment, the presentation of the figure (e.g. distance lines for electrostatic interactions) suggests more certainty than is warranted. Finally, note the pKA of inositol phosphates (Kooijman et al. 2009, PMID: 19725516); they will be partially protonated at pH 4.

We have revised Figure 3C to show the density of the lipid and important residues nearby. In our initial manuscript, we already acknowledged the fact that the assignment of PIP2 into the density is not unambiguous based on the cryo-EM map. However, the biochemical environment of the binding pocket, as well as the subsequent mutagenesis data support such a PIP2 binding site. We now re-wrote the paragraph to make this even clearer.

We thank the reviewer for pointing out the issue of PIO in the deposited model. We used the wrong model containing the head group, which is intended to be used only for making illustrations, for validation. We have now uploaded a new validation report using the correct model.

We acknowledge the referenced literature (Kooijman et al. 2009, PMID: 19725516) that the phosphate group may be partially protonated at pH 4, but this study is unable to reliably estimate the pK_a2_ of PI(4,5)P2 and the pK_a1_ is not studied. Moreover, in Figure 7 of the paper, no major difference is observed in the charge status of PI(4,5)P2 at pH 4-5, suggesting that the ionization status of PIP_2_ doesn’t change in the experimental condition investigated in our study. In a more recent review of PI(4,5)P2, it was suggested that at pH 4-5, both of the phosphomonoester groups are mono-protonated (Luís Borges-Araújo and Fabio Fernandes 2020, PMID: 32858905). Therefore, a net charge of -2 is present in the head group of PI(4,5)P2 under our experimental conditions, which allowed the molecule to interact favorably with the positively charged binding pocket (Figure 3C).